# Infrared Spectra of Small Radicals for Exoplanetary Spectroscopy: OH, NH, CN and CH: The State of Current Knowledge

**DOI:** 10.3390/molecules28083362

**Published:** 2023-04-11

**Authors:** Svatopluk Civiš, Adam Pastorek, Martin Ferus, Sergei N. Yurchenko, Noor-Ines Boudjema

**Affiliations:** 1J. Heyrovsky Institute of Physical Chemistry, Czech Academy of Sciences, Dolejškova 2155/3, 18200 Prague 8, Czech Republic; 2Department of Chemistry and Biochemistry, Old Dominion University, Norfolk, VA 23529, USA; 3Department of Physics and Astronomy, University College London, Gower Street, London WC1E 6BT, UK

**Keywords:** radicals, spectroscopy, infrared spectra, short living radicals, unstable species, atmospheric chemistry

## Abstract

In this study, we present a current state-of-the-art review of middle-to-near IR emission spectra of four simple astrophysically relevant molecular radicals—OH, NH, CN and CH. The spectra of these radicals were measured by means of time-resolved Fourier transform infrared spectroscopy in the 700–7500 cm^−1^ spectral range and with 0.07–0.02 cm^−1^ spectral resolution. The radicals were generated in a glow discharge of gaseous mixtures in a specially designed discharge cell. The spectra of short-lived radicals published here are of great importance, especially for the detailed knowledge and study of the composition of exoplanetary atmospheres in selected new planets. Today, with the help of the James Webb telescope and upcoming studies with the help of Plato and Ariel satellites, when the investigated spectral area is extended into the infrared spectral range, it means that detailed knowledge of the infrared spectra of not only stable molecules but also the spectra of short-lived radicals or ions, is indispensable. This paper follows a simple structure. Each radical is described in a separate chapter, starting with historical and actual theoretical background, continued by our experimental results and concluded by spectral line lists with assigned notation.

## 1. Introduction

For millennia, the existence of worlds beyond our solar system has been only a philosophical question. In 1917, Walter Sydney Adams obtained the very first indirect spectral evidence for the existence of an extrasolar planetary system by detecting calcium in the photosphere of white dwarf Wolf 28 (Van Maanen 2). Adams was not aware of the measurements indicating planetary debris and suggested that the spectral lines represented just a “pollution” in the observed stellar atmosphere. Almost a century later, his primacy was recognized by M. Zuckerman, who studied the archive observational data [1]. In 1988, Campbell et al. [2] pioneered a radial velocity survey in the first tentative detection of a real exoplanet named “Tadmor”,: a gas giant around Gamma Cephei A. The first exoplanet orbiting a main sequence star was then detected in 1995 [3].

Space telescopes and ground-based observatories have confirmed so far over 5300 worlds beyond the solar system. These instruments have shown an astonishing diversity in planetary physical properties and orbital parameters. However, less is directly known about their environments. In the upcoming era of space, telescopes such as JWST, Ariel and beyond [4] could fill this gap mainly by the observation of atmospheric spectral imprints [5]. Among various atmospheric components, an important role can also be played by unstable species typically observed in plasma, such as OH, NH, CN and CH.

Atmospheric composition and properties allow us to derive insights into the planetary formation and evolutionary processes, including specific spectral features documenting the impact history of early planets or random cataclysmic events in evolved planetary systems.

In the solar system, the scaling of the lunar crater density (the number of craters per lunar area) and the investigation of impacted melted fragments and volcanic rocks from Apollo and Luna missions indicate that the era of heavy bombardment persisted from the Earth’s origin and ended between 4.0 and 3.8 Gya [6]. The Moon–forming event, ca. 4.51–4.47 Gya, refs. [7,8] was followed by the solidification of the Earth’s crust within 2–10 Myr, terrestrial core-closure and probably by secondary collisions with large chondritic late-veneer objects [7,9]. These impacts are believed to significantly affect the atmospheres, hydrospheres, lithospheres and possibly upper mantles of planets, because of the impact of bodies lingering on unstable trajectories during the post-accretion phase of the solar system’s protoplanetary disk [10,11,12,13,14]. The chemistry that results from giant impacts is relevant for prebiotic chemistry and may have been essential for the emergence of life [15,16,17,18,19,20,21,22,23]. Direct records of the effect of giant impacts on the primordial environment on Earth and Venus have been erased by geological processes, such as plate tectonics (Earth) or global resurfacing (Venus) [24]. The earliest direct evidence of impacts on Earth is the 3.4 to 2.5 Gya old spherule layers in South Africa and Australia, and the oldest impact craters originate at 2.23–2 Gya [25,26]. Other planets have also experienced impacts. The cumulative number of impacts for Earth and Venus is typically assumed to be of the same order, but for the Moon, Mercury and Mars, the impact rate is thought to be one order of magnitude lower. Exact values differ in the current literature [6,27,28,29,30]. In the future, exoplanetary science should broaden this topic over unprecedented horizons regarding a wide range of planetary types as well as their possible chemical compositions, but fundamental questions such as if other planetary systems would have had the same impact history regarding mass delivery, impact frequency or the size distribution of impactors, has not been entirely resolved [7,31,32]. It can be expected that relevant data will be provided by the analysis of an exo-atmosphere composition.

The molecular imprints of impacts account for the delivery of simple species [33] such as CO, NH_3_, CO_2_, H_2_O, H_2_, H_2_S, SO_2_, HCN, HCHO, CH_3_OH, and maybe also HCONH_2_ [34] together with complicated substances such as polymers (tholins), ref. [35] various organic compounds and also prebiotic substances themselves [11,36,37,38]. These components originate from impactors [11,39,40] or result from plasma reprocessing during the hypervelocity atmospheric entry [22,41] and the final impact on the planetary surface [14]. However, a clear methodology for direct impact signature detection in the exoplanetary atmosphere is still a challenging problem for contemporary science [42].

Early exoplanetary atmospheres might have been very dusty due to volcanic activity and the ejecta of great impacts. This would have led to other events connected to the extensive occurrence of plasma: electric discharges [43,44,45,46]. Lightning is best studied in the Earth’s atmosphere but has also been discussed regarding solar system bodies and even exoplanets [11,43,44,45,47,48,49,50,51,52,53]. The first evidence of extra-terrestrial lightning was provided by Voyager 1 during its 1979 Jupiter flyby with the detection of electromagnetic waves—slowly falling tones at audible frequencies (whistlers) [54]. Subsequently, Jovian lightning was observed as localized flashes by onboard optical cameras [49,50,51,52,53]. Optical records indicate lightning on Earth, Jupiter [49] and Saturn [55], and electromagnetic measurements add Uranus and Neptune [56,57]. Electrical discharges are also suggested to exist in the atmosphere of Titan [58,59,60] and Venus [61,62] and transient luminous events (TLE) have been observed above thunderclouds on the Earth, ref. [63] suggested Jupiter [64] and predicted Saturn, Venus and Mars [65,66]. Martian dust storms are also expected to produce discharges [67]. Hodosán et al. [68] predicted that the optical detection of exoplanetary lightning (based on the example of HAT-P-11b) is limited by the orbital radius and optical power of the emission and concluded that a combination of radio and infrared observations might be required [68]. Lightning changes the atmospheric composition and spectral properties of a planet, as demonstrated by sulfuric and phosphoric condensates on Saturn or Venus, ref. [69] hydrocarbons and HCN in volcanic gases on early Mars, ref. [70] C_2_H_2_ and HCN on Jovian planets, ref. [71] NO production [72,73,74] and variations [75] on Earth or Venus, ref. [76] nitrogen fixation on early Earth, ref. [77] Mars [78,79] is the origin of hydrocarbons, nitriles, [80,81] surface organic deposition [82] and even the origin of prebiotic substances on Titan [17,83,84,85,86,87]. Regarding exoplanets, it is predicted that HCN spectral signatures will be significant for 2 to 3 years after its lightning production on exoplanet HAT-P-11b (as an example case). Hodosán et al. [88] also discussed the evidence for lightning activity on HD 189733b and deduced the expected lightning occurrence for exoplanetary prototypes. Ardaseva et al. [89] found that atmospheric electricity on an Earth-like exoplanet could substantially modify its spectral signature due to NO, NO_2_, O_3_, H_2_O and H_2_ and predicted that a significant abundance of cyanomethylidyne (C_2_N radical) and its chemical successors might serve as a lightning tracer in N_2_/CO_2_ atmospheres [89].

In the current study, we employed a wide range of experimental methods to explore simple radicals’ formation related to plasma processes in exoplanetary atmospheres by simulating three high energy scenarios:The shock wave and high energy plasma of impacting extraterrestrial bodies go into an early planetary atmosphere,Electric discharges of lightning in heavy clouds of dust, vapors and other aerosols are from impact, volcanic activity and evaporation in the early atmosphere,The ionization and fragmentation of stable molecules are present in planetary atmospheres due to the strong X-ray and UV radiation from the parent star.

We discuss spectral features of the most prominent species associated with the plasma reprocessing of planetary atmospheres: OH, NH, CN and CH. Non-local thermodynamic equilibrium (non-LTE) effects that often play an important role in such processes are illustrated. The description of spectral features summarized in this study provides data that are important for the future evaluation of their detectability, which, however, should take the mechanisms of atmospheric reprocessing, the parameters of an observed extrasolar system and observational strategy into account.

## 2. Instrumentation of the Spectral Measurements

A specially designed discharge tube was used in all measurements and experiments. Its scheme is depicted in Figure 1 below. This discharge cell represents only one of the possible experimental setups that are able to perform a spectroscopic study of various radicals. Nevertheless, the positive column of glow discharge inside the cell is relatively simple to realize and can be easily time-modulated by the switching of high voltage. These advantages predetermine the positive column as a very favorable choice for application in time-resolved FTIR spectroscopy.

The discharge cell is constructed from a temperature-resistant glass to endure numerous thermic shocks emerging from the pulsed glow high-voltage discharge. The anode and cathode are made from stainless steel. The cathode is hollow with a large inner area, which aims to lower electromagnetic noise, as was proven empirically. Both electrodes are physically connected by a glass tube of 12 mm in diameter and 25 cm in length, where the glow discharge emerges. The electric voltage between both electrodes is typically around 1 kV but slightly varies for each experiment. Both electrodes are encapsulated in double-layer glass, which allows them to be cooled by running water. The glass connecting tube is provided with an emission window at one side, which is chosen by actual experimental parameters. Usually, CaF_2_ or KBr windows are used. From the other side, the glass discharge tube is connected to a rotary vacuum pump. The whole system is, therefore, under a constant vacuum and the slight flow of a gaseous mixture. The vacuum in the discharge tube is typically around 2 × 10^−3^ Torr. The whole glass discharge tube is submerged in a cooling medium—usually water—also with dry ice or liquid nitrogen. An inert buffer gas, if used during the measurement, is supplied onto the cathode and anode at once. Additive gases (usually the measured ones) are supplied only onto the anode and drift through the positive column all the way to the cathode and then away from the discharge system due to constant evacuating. Emission radiation leaves the discharge cell through the emission window, which is positioned at the Brewster angle to prevent multiple reflections, and is optically-oriented to a lens, focusing the light beam to the entry aperture of the spectrometer itself.

An infrared spectrometer Bruker IFS 120 HR (Garching, Germany), specially modified for time-resolved measurements, was used in all experiments. A scheme of an inner arrangement of the spectrometer can be seen in Figure 2.

As can be seen in Figure 2, the heart of the spectrometer is the Michelson interferometer itself. An interference of signals together with the tracing HeNe laser signal (for the mobile mirror position tracing) is optically focused through the sample space (depicted in Figure 2 as a “cell”), where a sample can be put into the path of a light beam (to obtain absorption spectra) and finally led to the detector space. InSb and MCT (HgCdTe) semiconductor detectors were used in all experiments. Both detectors are cooled by liquid nitrogen; an InSb detector operates in the 1800–13,000 cm^−1^ spectral range and MCT in the 700–3000 cm^−1^ range.

### Time Resolution

The time-resolved experimental system used is the key feature of our measurements, differentiating it from common infrared spectroscopy. Adding a third dimension (time) into the spectra allows not only to observe the kinetics of selected spectral transitions but also to average specific time-shifted spectra, thus improving the signal-to-noise ratio. In addition, the typical problem when spectra of different species overlap and cover each other can be efficiently resolved by measuring time-dependent spectra. Indeed, many different rotation-vibration-electronic states differ in lifetimes, especially unstable ones, which helps separate them in time. The principle of the time-resolved measurements used is depicted in Figure 3.

The position of the mobile mirror is deduced from the HeNe laser fringes, which can be seen in the top layer of Figure 3. This laser signal has an original shape of a cosine function but resembles a rectangular shape after digitalization. The frequency of this rectangular signal depends on the speed of the mobile mirror. Typically, the mobile mirror speed is set to an adequate frequency of 10 kHz (speed is typically expressed in the units of frequency by Bruker company), which then defines the width of one laser signal to 100 µs. The whole process consisting of one measurement and data acquisition is realized in these 100 µs. The discharge pulse is controlled by a high-voltage switch (Behlke electronic GmbH, Frankfurt, Germany), and its width (duration) can also be set to various values. A typical pulse width in the experiments performed is 22 µs, which is also depicted in the middle layer of Figure 3. Continuous data acquisition is possible during the whole length of the discharge pulse and after it. The acquisition is controlled by an AD switch (“Data acquisition trigger” in Figure 3), which can also be pre-programmed to obtain various numbers of data points (typically 30, maximally 64) and their acquisition time intervals (every third microsecond in Figure 3).

In this paper, spectra between 30 and several hundred scans, depending on the sample, were coadded to obtain a reasonable signal-to-noise ratio.

## 3. OH Radical

### 3.1. Theoretical Background

The OH radical is a short-living intermediate, usually accompanying processes of burning and explosion. It is known that OH plays a key role in the Earth’s atmosphere, acting as the main species in oxidation reactions in the troposphere. OH can be found in solar spectra, in interstellar diffuse clouds and in planetary atmospheres.

The potential curves of the OH radical in different energy states are illustrated in Figure 4.

It is almost a century ago since the first information on the spectra of OH was published [92]. This work contained the very first clear detection of an OH radical, usually identified as one of many water bands up to this time. A lot of publications have followed this evidence of the spectral detection of OH, which initially focused mainly on experimental spectra in a wide spectral range, but later also aimed at a more theoretical approach, allowing a more precise description of quantum processes on a molecular level.

In 1950 I. Meinel found emission lines of the OH radical in the UV spectra of the night sky [93], which he used to empirically determine the rotational temperature of OH to be 260 ± 5 K. Later, it was discovered that the origin of these lines was the ground electronic state of OH X^2^Π. These “Meinel bands” (or so-called night-time airglow) occur in the reaction of hydrogen atoms with the atmospheric ozone in the upper mesosphere approximately 90 km above the surface of the Earth. These bands serve as an important tool for the quantification of oxygen abundance in the Sun’s chromosphere and photosphere (as well as in other stars). Recently, fluctuations of Meinel bands in far infrared regions were examined in order to prove the effect of gravitational waves [94]. Many in situ measurements of Meinel bands were performed, but the first extensive work to faithfully imitate the conditions in the Earth’s mesosphere was a study by Abrams et al. (1994) [95], who measured Meinel bands with an artificial light source in the laboratory in the 1850–9000 cm^−1^ spectral range. One of the newest theoretical explanations on Meinel bands and the dependent role of OH radicals was described by Chen et al. [96].

Purely experimental studies have focused mainly on the measurement of the maximal number of OH lines that consisted of detection in photographic films, ref. [91] the flash photolysis of hydrogen peroxide vapors, ref. [97] the reaction of NO_2_ with hydrogen in a flow cell, ref. [98] an oxygen-acetylene flame to simulate hotter radiation sources [99] or even double laser resonance for the specific excitation of OH to required energy levels [100]. Additionally, laser-induced fluorescence [101] or a simple emission measurement of OH in the electric discharge of a helium–water mixture [102] was not an exception.

K. R. German was one of the first who succeeded in the experimental measurement of the radiative lifetime of the OH radical in its first excited state A^2^Σ^+^, setting the value to 0.688 ± 0.021 μs [103]. This value was theoretically supported by C. W. Bauslicher in 1987 [104].

Experimental data, including the positions and intensities of OH lines, also led to the theoretical prediction of other energy levels of OH radical and to the calculation of molecular constants allowing the modeling of the OH spectrum in the whole spectral region [105,106,107,108]. Many publications were also focused on the calculation of molecular parameters and transition moments [109,110,111,112].

The detection of OH in solar spectra is considered to be one of the most important events, given that the pure rotational lines of OH can be used as a tool for the Sun’s oxygen abundance calculation with high precision and almost independently of other models and variables. The first solar oxygen abundances estimated by pure rotation OH lines were published in Goldman et al. (1983) [113], where the possibility of the use of CH rotational transitions for the calculation of the solar oxygen-carbon ratio was also discussed. Grevesse et al. [114] focused on rotation-vibration solar spectra and the emerging possibilities of oxygen quantification. Rotational transitions in the ground level, first and second vibrationally excited states of OH, and also rotation-vibration transitions with high *J* numbers were found in the solar spectra [115]. One of the newer publications on solar oxygen abundance was the work of Asplund et al. [116], who focused on the implementation of modern 3D hydrodynamic models of the solar atmosphere.

Part of the newest publications, using well-examined OH spectroscopy [117,118,119,120], usually also focus on the detection of the OH radical in exoplanetary atmospheres or stars. In such cases, the OH radical again serves as a tool for oxygen abundance calculation on metal-poor stars [121] or for the understanding of radiative transitions in the atmosphere of terrestrial planets, e.g., Venus [122]. Recently, the OH radical was detected in the atmosphere of the WASP-33b exoplanet, ref. [123] which is the very first detection of hydroxyl radicals in an exoplanetary atmosphere.

### 3.2. Experimental Results

The rotational and rotational-vibrational spectra of OH were first examined in the glow discharge of water vapor mixed with a helium buffer. This initial mixture was, however, proven ineffective, and a mixture of hydrogen and oxygen in helium was used instead. It was also empirically proven that a combination of partial pressures of 1 Torr of helium, 0.6 Torr of hydrogen and 0.6 Torr of oxygen produced generally the best spectra. The electric voltage across both electrodes in this glow discharge was 1.4 kV, and electric the current was 100–150 mA. MCT and InSb detectors, as well as corresponding optics, were used. Spectral resolution varied between 0.02 and 0.05 cm^−1^ and the entry aperture of the spectrometer was set to 2 mm (4 mm when MCT was used). The pulse width was 22 µs, the offset was set to 0 µs and data acquisition was performed at each second µs. The number of data acquisition points was 30; therefore, a 60 µs time window was covered by the measurement. The number of scans used for accumulation, and the improvement of signal-to-noise ratio varied between 100 and 1000, 400 and 1000 cm^−1^ and 1800 and 3500 cm^−1^ interference optical filters were used.

The rotational-vibrational spectrum of OH radical in the X^2^Π ground electronic state is depicted in Figure 5.

The spectrum in Figure 5 contains strong emission atomic lines of helium and molecular lines of water with an intensity maximum of around 3500 cm^−1^, covering the range up to 2500 cm^−1^. This spectrum was recorded with 0.05 cm^−1^ spectral resolution and the accumulation of 1000 scans. Spectra between 20 and 32 µs times were selected for averaging due to the best intensity.

The overall rotation spectrum of the OH radical for the 700–1000 cm^−1^ spectral range is depicted in Figure 6.

Figure 6 captures the rotational spectrum of the OH radical in the ground and the first vibrationally excited state. A total of 300 scans at the resolution of 0.02 cm^−1^ were accumulated to obtain this spectrum. Spectra between 16 and 28 µs times were averaged to obtain a reasonable signal-to-noise ratio.

The Appendix A provides all rotational-vibrational emission lines of the OH radical for the 1800–3500 cm^−1^ spectral range obtained in this study. The lines were assigned by the use of the HITRAN database, which was then used in the spectral calibration. This emission spectrum of OH belongs to the X–X transition type (rovibrational transitions in the ground electronic state), which also undergoes splitting by the influence of the spin–orbital interaction, orienting the overall spin angular momentum to be parallel or antiparallel. OH can, therefore, exist in two ground electronic states—X_1/2_ or X_3/2_. The OH radical in the X_3/2_ state undergoes Λ-doubling as well.

Apart from the rovibrational spectra, the Appendix A also lists all the observed pure rotational lines of the OH radical in the 700–1000 cm^−1^ spectral range. The pure rotational (Δ*v* = 0) emission lines were gathered into quadruplets with an exactly defined rotational constant, determining the distance between individual quadruplets. The selection of a 700–1000 cm^−1^ spectral range was due to the limited sensitivity of the used MCT detector for wavenumbers below 700 cm^−1^. The data acquisition in the range around 1000 cm^−1^ and above was also complicated due to high *J* numbers, requiring a relatively hot radiation source.

The rotational transitions reported here are the first experimentally laboratory-observed pure rotational OH lines obtained by use of a cold radiation source. However, it must be noted that these data are already known from solar spectra (ACE and ATMOS) and are published.

### 3.3. Comparison with Solar Spectra

The experimental data of the OH radical were compared with the ACE [124] solar spectra. Figure 7 illustrates a comparison of the experimental OH emission lines with the ACE solar spectra in the 2880–2910 cm^−1^ spectral range.

Green highlighted strips in Figure 7 show the match between the experimental and solar spectrum. As expected, the distribution of emission intensities was different for both spectra due to the different temperatures of the corresponding energy sources.

Figure 8 depicts the comparison of the solar ACE spectrum with the experimental OH lines in the 700–1000 cm^−1^ spectral range (pure rotational transitions).

As mentioned above, green highlighted strips show the coincidences of the spectra. It can be easily seen that solar spectra are highly rotationally excited, with transitions even around and above 1000 cm^−1^. This is caused by higher temperatures in the Sun in comparison to our discharge experiment.

### 3.4. Theoretical Comparison Using a Non-LTE Model

Here, we applied the MoLList for OH to model the experimental time-dependent spectrum. The intensities of the vibrational bands 4–3, 3–2, 2–1 and 1–0 indicate a departure from the local thermal equilibrium (LTE). The intensities of the rotational lines within individual bands can be described by a single rotational temperature of about 500 K at any time step.

In order to obtain the vibrational temperature and its possible time dependence, we selected a set of the strongest lines from each of the four bands and estimated their intensities: l69 lines from *v* = 1 to 0, 56 from 2 to 1, 27 from 3 to 2, 21 from 3 to 2 and 7 from 4 to 3. Since the experimental intensities were not calibrated, we used theoretical absolute intensities to relate intensities between different vibrational bands through a relative vibrational state population $n_{v}$ as follows. We first introduced an emission line intensity for a given rotational temperature *T*_R_ and an upper stare vibrational population $n_{v}$ as given by:(1)$$I\left(i\to f\right)=g_{J_{i}}\frac{n_{v}e^{-c_{2}{Ẽ}_{i}^{v,\mathrm{rot}}/T_{R}}}{Q_{R}\;\left(T_{R}\right)}A\left(i\to f\right),\;$$
where $c_{2}=hc/k_{B}$ is the second radiation constant, Ẽ=E/hc is the energy term value, $g_{J_{i}}=g_{}^{ns}\left(2J_{i}+1\right)$ is the state degeneracy, $g_{}^{ns}$ is the nuclear-spin statistical weight factor and $T_{R}$ is the rotation temperature. $Q_{R}\left(T\right)$ is a rotational partition function defined as:(2)$$Q_{R}\left(T\right)=g_{}^{ns}\underset{n}{\sum }\left(2J_{n}+1\right)e^{-c_{2}{Ẽ}_{n}^{v,\mathrm{rot}}/T_{R}},\;$$
where $A_{if}$ is the Einstein-*A* coefficient and $I_{if}$ is in photon/s for a transition i→f with the wavenumber ${\tilde{\nu }}_{if}={Ẽ}_{J,\nu }^{i}-{Ẽ}_{J,\nu }^{f}$. The total energy is approximated by a sum of the rotational and vibrational energies:(3)$${Ẽ}_{J,\nu }={Ẽ}_{\nu }^{vib}+\left({Ẽ}_{J,\nu }-{Ẽ}_{\nu }^{vib}\right)$$
where v and k are the vibrational and rotational quantum numbers, respectively, and ${Ẽ}_{J}^{\nu ,rot}={Ẽ}_{J,\nu }-{Ẽ}_{\nu }^{vib}$ is the rotational energy contribution.

Assuming the Boltzmann equilibrium for the vibrational degrees of freedom at $T_{\mathrm{vib}}$, the vibrational population $n_{v}$ of state v is given by:(4)$$n_{v}=\frac{e^{-\frac{c_{2}E_{v}^{vib}}{T}}}{Q_{vib}\left(T_{\mathrm{vib}}\right)}$$
where $Q_{\mathrm{vib}}\left(T_{\mathrm{vib}}\right)$ is a vibrational partition function defined as:(5)$$Q_{\mathrm{vib}}\left(T_{\mathrm{vib}}\right)=\underset{v}{\sum }e^{-c_{2}{Ẽ}_{v}^{vib}/T_{\mathrm{vib}}}.$$

We applied the bi-temperarure intensity expression to model the experimental spectra of OH in Figure 5 using the empirical line list MoLLIST. By varying the rotational and vibrational temperatures, we obtained $T_{\mathrm{rot}}\approx$ 500 K and tentatively estimated $T_{\mathrm{vib}}\approx$ 3900 K. We used the ExoCross program to simulate the molecular spectra of OH.

An example of MoLLIST theoretical spectral lines modeled with $T_{\mathrm{rot}}=$ 500 K and $T_{\mathrm{vib}}=$ 3620 K is shown in Figure 9. As shown in this figure, the theoretical rotational lines reproduced the experimental pattern well.

Assuming that the vibrational population $n_{v}$ is uniform across all rotational states with a given (upper) vibrational state and keeping in mind Equation (1), the vibrational number density can be estimated as a ratio between the experimental and theoretical line intensity:(6)$$n_{v}=\frac{I^{\exp}\left(i\to f\right)}{I_{v}^{\mathrm{calc}}\left(i\to f\right)},$$
where
(7)$$I_{v}^{\mathrm{calc}}\left(i\to f\right)=g_{J_{i}}\frac{n_{v}e^{-c_{2}{Ẽ}_{i}^{v,\mathrm{rot}}/T_{R}}}{Q_{R}\;\left(T_{R}\right)}A\left(i\to f\right),$$

Here, we used $T_{R}=$ 500 K in conjunction with the MoLLIST line list for OH and obtained the number densities for $v^{\prime }=1$, $v^{\prime }=2$, $v^{\prime }=3$ and $v^{\prime }=4$ by averaging Equation (6) over selected rotational transitions within the corresponing bands ($v^{\prime }$,$v^{\prime }-1$):(8)$$n_{v}=\underset{i\to f}{\sum }\frac{I^{\exp}\left(i\to f\right)}{I_{v}^{\mathrm{calc}}\left(i\to f\right)}\;$$

The experimental data were recorded every 2 microseconds after the pulse for a total of 54 µm, producing spectra for 17-time steps. The analysis was performed for (1,0), (2,1), (3,2) and (4,3) and focused primarily on the outputted spectra for nine of these time steps, at 2, 8, 16, 24, 30, 36, 42, 48, and 54 µs. Namely, the strongest emission lines for each band were extracted and fitted using a Lorentzian profile fitting tool via the data analysis software OriginPro 2021 to determine the intensity of the peak. This was conducted for a total of 10, 6, and 8 strongest lines of the (1,0), (2,1) and (3,2) bands, respectively. The obtained magnitude was equated to the intensity of the theoretical emission line to obtain the corresponding number density $n_{v}$ as a scaling factor and as a measure of the relative intensity of the experimental spectra with respect to the theoretical spectra. As the intensity of an emission line is dependent on the number of molecules occupying a certain band, this factor also gives an indication of the number of molecules.

The theoretical spectra $I_{v}^{\mathrm{calc}}\left(i\to f\right)$ were computed using Equation (1) with the vibrational population $n_{v}$ normalized to 1, i.e., $n_{v}=1$ and are illustrated in Figure 10.

Note that the experimental intensities were not calibrated $I^{\exp}\left(i\to f\right)$ to the absolute measure; therefore, the $n_{v}$ values in Equation (8) represent relative populations.

The vibrational populations of OH as a function of time were estimated using Equation (6) and are illustrated in Figure 11, showing the general trend of the OH molecule number density in the discharge experiment.

We can now take a ratio of these populations to each other. In Figure 12, the populations of OH for v=2 and v=3 relative to the population of v=1 are shown. The non-LTE relative populations of v=2 approximately agree with the vibrational temperature of $T_{\mathrm{vib}}=$ 3900 K, estimated above, while the v = 3 state population appears to be higher but within the estimated error. We aslo observed a small trend for the excited states, which were depopulated by the end of the experimental cycle after 30 µs.

All measured transitions of OH are summarized in Appendix A of this article.

## 4. NH Radical

### 4.1. Theoretical Background

The NH radical is a diatomic molecular fragment that is readily formed in high-energy and decay events of nitrogen and hydrogen-containing molecules. NH can be easily spectrally detected in flames of such compounds. In astrochemistry, NH serves as a tool for nitrogen abundance calculation, especially in stars’ gas envelopes and interstellar dust clouds. The quantification of nitrogen by the use of the NH spectra has one important advantage—the calculation is independent of the contribution from other elements. For example, in the case of nitrogen abundance calculation through the use of the CN radical, carbon contribution must not be omitted, which brings additional complexity into the whole estimation process.

Potential curves for low-lying energy levels of NH radical are depicted in Figure 13.

The first spectral detection of the NH radical is dated to 1893 by J. M. Eder [126] through the use of photographic detection. In 1919 Fowler et al. [127] published a study where he described the detection of the same NH bands in solar spectra.

An extensive study on NH spectra was, however, first published in 1936 by G. Funke [128]. This study focused on absorption spectra in the UV spectral range around 336 nm. Additionally, here, sensitive photographic detection by use of a grating spectrometer was used.

The discovery of the NH radical in the solar spectra increased interest in the detailed understanding of NH spectral features and new extensive publications on the fundamental spectroscopy of NH, which were produced and measured by cold laboratory light sources and began to emerge quickly. NH was studied in many experimental ways, namely, e.g., using the flash photolysis of isocyanic acid (HNCO), ref. [129] the photolysis of ammonia mixed with inert gas, ref. [130] UV photolysis of ammonia followed by the resonant fluorescence of NH, ref. [131] the electric discharge of nitrogen–hydrogen mixture in a hollow cathode [132] or modern two-photon capture on ammonia in a flow reactor through the use of a tunable laser [133]. Ram et al. [134] later also discussed the effectivity of UV/VIS detection on the NH radical by cryogenic echelle spectrograph in comparison to the classical FTIR spectrometer, when it was proven that a cryogenic spectrograph in Kitt Peak National Observatory in Arizona (so-called Phoenix) was more sensitive.

From an astronomical point of view, the spectra of the NH radical are mostly used in nitrogen abundance calculations for stars and interstellar dust clouds. NH was also detected in a cometary environment, e.g., the detection of NH in the tail of comet Cunningham [135], together with CN, OH and CH, can be mentioned. The NH radical detected in comets can also be used for the calculation of photo-dissociative lifetimes of specific transitions, as was demonstrated in the case of the A^3^Π → X^3^Σ^−^ transition by P. Singh in his work [136]. NH can be found in cold stars, e.g., HD 122563, ref. [137] which is also a metal-poor star. Rovibronic bands of NH can also be spotted on Betelgeuse [138]: the second-brightest star of the Orion constellation. V. Smith et al. [139] found NH bands in the spectra of 12 red giants in 1986 and used them to estimate the abundance of C, N, O and other elements. A similar paper by Aoki et al. [140] studied the high-resolution IR spectra of CN and NH from K and M class giants. Aoki pointed out the advantage of nitrogen abundance estimation through the use of NH in comparison to CN.

The first evidence of the existence of NH in interstellar space was published in the work of D. Meyer in 1991, ref. [141] where its presence was proven in the diffuse cloud in the proximity of Zeta Persei and HD 27778. The spectra of NH were measured in the UV region and assigned to the well-known A^3^Π → X^3^Σ band. In 1997, the NH radical was also detected in an interstellar cloud close to Zeta Ophiuchi: a bright star approx. 366 lightyears from Earth [142]. Data obtained during this measurement effectively disproved the hypothesis of the so-called hot formation of NH, when NH was formed by the direct synthesis of nitrogen and hydrogen and a subsequent formation of atomic hydrogen. Instead, the data supported the theory of the chemical reactions’ dominance on the surface of dust grains in diffuse molecular clouds as the main source of interstellar NH. This theory was also supported by Weselak et al. in 2009 [143].

A possible complex origin of nitrogen in metal-poor stars was discussed in the work of Spite et al. [144] in 2005. Spite and his team focused on the explanation of the formation of light elements in the early ages of our galaxy through the use of the calculation of abundance ratios of various elements and by modeling their mixing when a significant star layer mixing took place. The publication compared data collected from 35 different stars and concluded that the gas composition on the surface of a star that did not undergo any layer mixing procedure was similar to the composition of these gases during the star’s formation in the original ratio.

In 1970, a work by Claxton [145] was published, describing the ab initio calculation of the NH radical bond length, together with the determination of preferential formation of NH ions in (NH_4_)_2_HPO_4_ crystals irradiated by gamma photons. A detailed analysis of NH spectra through the use of Dunham fit was described in G. Das et al. [146], showing that only five parameters were needed to construct the whole theoretical NH spectrum. Ram et al. [147] revised the molecular constants of NH in 2010, giving more precise values for the same theoretical construction of NH spectra. Many of these publications, focusing on precise molecular parameters for the theoretical modeling of the NH spectra, are constantly published up to this date [125,148,149].

Nowadays, new applications of the NH radical are emerging in the field of magnetic molecular traps [150]. Such traps are used to study the physical properties of ultra-cold and very high-density matter. The NH radical, having just the right properties for this application, was decelerated to extremely low kinetic energy, thus cooling itself to very low temperatures. For this application, a specific energy state of NH, namely metastable a^1^∆, was excited to the A^3^Π state and subsequently deexcited to the ground state X^3^Σ^−^. The deceleration of NH has been realized via the Stark [151] or Zeeman [152] decelerator. These experiments, leading to the ideally kinetically decelerated NH radicals of high local density, have high potential importance in quantum computers.

### 4.2. Experimental Results

The discharge was realized either with a mixture of hydrogen and nitrogen (pressure ratio 1:1, the total pressure of 3 Torr) or with a mixture of nitrogen and ammonia in an argon buffer gas. In the latter, the pressure was 0.5 Torr of argon, 0.8 Torr of nitrogen and 0.1 Torr of ammonia. The spectral resolution was 0.02 cm^−1^. Electric voltage varied between 1 and 1.4 kV, and electric current was 150–300 mA. The aperture of the spectrometer was set to 2 mm.

Figure 14 demonstrates the overall spectra of NH radical in the 2400–3600 cm^−1^ spectral range.

In panel A of Figure 14, the spectrum of the H_2_ + N_2_ discharge, which averaged over the most intense times, is displayed. This spectrum was obtained by collecting 50 scans. The bands are marked according to vibrational transitions. It is possible to notice that the second mixture (Panel B, NH_3_ + N_2_ + Ar) produced a spectrum with a higher signal-to-noise ratio, and the population of rotational-vibrational bands was slightly different as well. It must be mentioned that the second spectrum was measured by the collection of 100 scans.

The observed emission lines of the NH radical in the X^3^Σ^−^ the ground state are provided in the Appendix A. The formatting convention is the same as in the former case of the OH radical.

It is possible to observe the mutually inverse effect of the emission intensity distribution between nitrogen and the NH radical in the N_2_ + NH_3_ + Ar mixture. Figure 15 below demonstrates the time profiles of selected NH 1-0 and excited nitrogen W^3^Δ_u_-B^3^Π_g_ emission lines (3439.437 and 2676.777 cm^−1^) and their time-resolved spectra.

According to the time profile curves in Figure 15, it is possible to observe that both species gain energy in the discharge pulse (30 μs here). After the pulse, nitrogen starts to lose its energy faster than the NH radical, which lives slightly longer as a result of possible mutual energy exchange. At the 28th microsecond, which is still in the discharge pulse, the overall spectrum (in the red color) is depicted in the right top corner of Figure 15. This spectrum is completely occupied by nitrogen, and it is difficult to distinguish any NH radical lines here. The right bottom corner of Figure 15 demonstrates the spectrum (in blue color) of the 46th microsecond of the measurement. As can be seen, the spectrum contains no residual nitrogen bands, and only the NH radical is present. It is very difficult to distinguish NH from nitrogen by the use of non-time-resolved FTIR spectroscopy since nitrogen covers all other lines due to its omnipresent nature in the middle IR region. Nevertheless, by the application of time resolution, both species are separated in time and are, therefore, easily recognized in their pure form.

Through the use of time-resolved FTIR spectroscopy, we were also able to measure several pure rotational lines of NH (v = 0). According to our best knowledge, this measurement is one of the first experimentally observed pure rotational NH lines in the laboratory by the use of a cold source.

As can be seen in Figure 16, pure rotational lines of NH radical are situated in the range of approx. 700–900 cm^−1^ and are resembled in triplets. Our spectral resolution, 0.02 cm^−1^, was not enough to separate individual peaks, and their approximate position had to be estimated by the deconvolution of the whole triplet.

Our observed pure rotational lines of NH radical are summarized in Table 1 below.

With respect to ACE (Atmospheric Chemistry Experiment) data, we compared our NH radical spectra with ACE solar spectra. The match of our and ACE NH lines can be observed in Figure 17.

Figure 17 demonstrates a comparison of our NH radical experimental spectrum with the ACE solar spectrum. The coincidences of matching NH lines are highlighted in green color. The lines depicted in Figure 17 correspond to the 1-0 band of NH.

Figure 18 depicts the comparison of the ACE solar spectrum with our NH radical pure rotational triplet lines. The coincidence of the solar spectrum is highlighted by the green color, but it is also possible to see that the solar spectrum has a low population of pure rotational NH lines which are not resolved into triplets. For the information on plasma temperatures, the vibrational temperature has been calculated from the rotational-vibrational NH lines in the 2700–3500 cm^−1^ range for the N_2_ + NH_3_ + Ar mixture. An estimation of the vibrational temperature is given in the Appendix A of this article.

### 4.3. Theoretical Comparison with a Non-LTE Model

An LTE emission spectrum of NH obtained from the NH_3_ + N_2_ + Ar mixture was computed using the ExoMol line list MoLLIST [156,157] and is shown in Figure 19 (bottom) compared to the experimental spectra (top) for the 2500–3500 cm^−1^ range.

Calculations were performed in emission with the LTE temperature *T*_V_ = 6000 K, which provided the best reproduction of the experimental spectrum.

A non-LTE emission spectrum of NH obtained from the H_2_ + N_2_ + He mixture was computed for the 2400–3600 cm^−1^ range and is shown in comparison with the experimental spectra in Figure 20. Calculations were performed in emission using the ExoMol line list MoLLIST [156,157]. The best agreement with the experiment was achieved for $T_{R}$ = 500 K and $T_{v}$ = 8000 K as the rotational and vibrational temperatures, respectively.

## 5. CN Radical

### 5.1. Theoretical Background

The radical CN, together with the most stable molecules such as CO and HCN, were detected in all of the studies of the C, H, O and N-containing mixtures. In the reaction mixtures such as CO + NH_3_ or H_2_O + NH_3_, we have observed CH and NH radicals among other formamide decomposition discharge products. These radicals play a very important role in plasma chemical reactions [17,19,158]. However, the rovibronic transitions of the CN radical and the rovibrational transitions of a vibrationally excited CO molecule were dominant in the emission spectra of the studied formamide mixtures. The CN radical, due to its very rigid structure (strong triple bond) and a rather complicated electronic and rovibrational spectrum, allowed transitions between a whole series of excited states, such as A^2^Π − X^2^Σ^+^ Δv = 2, 3; the ground state X^2^Σ^+^; violet system B^1^Σ^+^_u_–X^2^Σ^+^_g_ in laser-induced plasma; etc. The first eight states of CN are shown schematically in Figure 21. Potential energy curves have been schematically plotted using constants taken from Ref. [159]. Due to very favorable Franck-Condon factors, the CN radical could be excited to very high vibrational and rotational states, which result in observations of a vast range of transitions from the visible spectrum through the infrared to the microwave regions [18,158,160,161,162,163,164]. Together with this capability, this reactive species exhibits high stability due to the bond dissociation energy of 7.77 eV (62,711 cm^−1^) [165]. The CN radical was thus capable of absorbing a huge amount of energy and populating a great number of energy levels without dissociating. Due to this effect, this species could interact (via energy transfer) practically with any molecule or radical in a broad spectral range from the ultraviolet up to the microwave region. The resilience of this radical and its common acquisition of a hydrogen atom to generate HCN meant that the latter species was an inherently formed product of high energy chemistry in C, H, O and N-containing atmospheres. The CN radical and CO molecule is unique in being able to store the thermal energy that would otherwise be dissipated in a multi-step reaction via the excitation of their numerous vibrational and rotational transitions. They, thereby, control the energy flow through the whole reaction pathway.

CN is also an important diatomic radical from the point of view of fundamental spectroscopy, which can be easily spectrally detected in high-temperature events such as burning, explosions, discharges, or exothermic chemical reactions and even on stars and in interstellar space. We should also highlight the huge importance of the CN radical for prebiotic chemistry, where it plays a key role as an energy mediator and allows the smooth performance of a vast number of prebiotic syntheses by the donation of excitation energy stored in countless electronic, rotational-vibrational and even pure rotational energy levels.

From the point of view of fundamental spectroscopy, CN radical spectra can be divided into many groups according to various parameters, but historically, the astrophysical division was found to be the most used. First, the system belongs to A^2^Π → X^2^Σ^+^ transitions, which fall into the wide spectral range of a red part of visible to middle IR. This system is frequently referred to as “red” for its lower wavenumber region. The second system referred to as “violet”, encompasses all transitions within principal B^2^Σ^+^ → X^2^Σ^+^ transitions, which are positioned between the visible and UV spectral region.

One of the first works that focused on deeper spectral analysis of CN radical was the paper by Francis A. Jenkins from 1928 [167]. Jenkins broadened the list of the violet CN system, called “tail bands”, according to their spectral band shape, which was first recognized by T. Thiele in 1897 [168]. As usual, the detection was realized by the use of a photographic film and a classical grating spectrometer.

Rotational-vibrational transitions of CN in the X^2^Σ^+^ ground state, located around a 5 µm wavelength (2000 cm^−1^), were first measured in 1975 by R. Treffers [169]. He detected three lines of the CN radical through the use of the King’s furnace, with emitting radiation coming from the heating of nitrogen and carbon monoxide in an air mixture. In 1978, D. Cerny et al. [170] performed an extensive spectral analysis of the red system of CN using the FTIR spectrometer. The measurement was realized in the 4000–11,000 cm^−1^ spectral range, and the CN radical was generated in the flame of N_2_O and acetylene. This publication also gave one of the first global fits of the CN spectra. Measurements by Cerny et al. were supplied by another laboratory data of CN in the ground state in 1982 when Davies et al. studied CN formed in the electric discharge of nitrogen mixed with cyanogen [171]. In our previously published papers, we analyzed the high-resolution electronic and rotation-vibration transitions of CN in a spectral range of 1800–3000 cm^−1^; see Horká et al. [160] and Civiš et al. [161]. The millimeter and sub-millimeter (microwave) spectra of CN radical were studied by D. Skatrud et al. [172], who measured 65 new lines in the microwave and FIR spectral regions (220–453 GHz). The experiment was realized in the electric discharge of nitrogen and methane. The millimeter spectra of CN were also studied by Bogey et al. [173] in 1986, who broadened the referential data for CN in this region up to v = 9. Unlike Skatrud, Bogey generated a CN radical in a radiofrequency discharge of CO + N_2_ + He mixture at the liquid nitrogen temperature. It is also worth mentioning the work of Ito et al. (1991) [174] on the microwave spectroscopy of CN, who provided high-quality CN spectral data and broadened the line list of transitions up to v = 10.

Prasad et al. in 1992 focused on the method of a supersonic expansion of the CN radical [175]. His specific experimental technique, when the CN was generated in a coronal discharge of a helium-methylazide-diazoacetonitrile mixture, was subsequently cooled by supersonic speed blasting, which allowed him to precisely measure the violet (B^2^Σ^+^ → X^2^Σ^+^) CN system for Δ*v* = 0 and Δv = 1. A study by Rehfuss et al. [176] was also very similar.

Wurfel et al. [177] measured the infrared fluorescence of CN in 1993 trapped in the solid crystal matrices of noble gases. This experimental arrangement allowed the study of trapped CN for a relatively long time, only proving the importance of the spectroscopy of the target molecules trapped in solid matrices.

A newer study by Liu et al. [178] focused on a rotational-vibrational structure of v = 2 → 0 in the red CN system using the concentration modulation laser spectroscopy; a work by Chao-Xiong et al. [179] focused on a very similar experimental technique to study the CN radical in a narrow visible region of 17,450–17,830 cm^−1^ has to be mentioned as well. Other publications to mention are Horká et al. (2003) [160], Hempel et al. (2003) [180], Li et al. (2003) [181], Hübner et al. (2005) [182] and Civiš et al. (2008) [161].

Many exclusively theoretical works have been published for the CN radical as well. For example, Ito et al. [183] focused on the perturbation analysis of emission spectra in the violet CN system for higher vibrational quantum numbers (up to v = 17) and later, even for v up to 19 [184]. Ram et al. [185] focused on the FTIR emission spectroscopy of CN in the red system and obtained a set of new and precise spectral constants. In the same year, Ram et al. [186] published an extension of his spectral parameters for the red CN system by measuring in the 3500–22,000 cm^−1^ spectral region. Another work by the same author was published in 2012, ref. [187] when yet another set of spectral constants was made more precise. Einstein coefficients for both systems of CN were studied by many authors, e.g., Brooke et al. [188] in 2014 is among the newer ones.

In 1994, Pradhan et al. [189] focused on the study of the dissociation energy of CN, where he obtained a value of *D*_0_ = 7.72 ± 0.04 eV using theoretical modeling. This value is obviously exceptionally high due to the triple covalent bond. This dissociation energy allows CN to exist in many excited states without breaking the bond.

The CN radical can also be generated in a DC discharge of methyl thiocyanate (CH_3_SCN). This means of generation was extensively studied by Li et al. in 2004, ref. [190] who found that lower concentrations of CN were achieved when a higher plasmatic flow was applied. On the contrary, a concentration of unstable CS rose at the same time. These findings were applied in the industry of thin-layer semiconductors.

Yet, another way in which CN radical generation is the photolysis of simple molecular gases was studied in 2011 when Hodyss et al. [191] tried to simulate the environment of Triton (the coldest body in the Solar system) and Pluto using thin cryogenic films of methane, nitrogen and carbon monoxide ices in specific ratios. A large variety of products were obtained by the UV bombardment of such ices, e.g., ethane, acetylene, HCO or CN radical.

CN can be found in the stars’ envelopes and in interstellar space. Its spectra are used to determine the C/N ratio and their isotopic abundance, leading to the possibility of star evolution modeling. The application of such data was well discussed in the work of Lambert et al. [192] from 1974, where the isotopic ratio of ^12^C and ^13^C on Betelgeuse was studied. A final ratio here was found to be ^12^C/^13^C = 7.0 ± 1.5, which showed a significant dominance of ^12^C. One of the very first publications on CN in interstellar space was the work by Saleck et al. (1993) [193], where the CN radical was located in the Orion A constellation. The discovery of Saleck et al. was supported by their own observation of the same system by the KOSMA telescope one year later [194]. Sneden et al. [195] published a comprehensive database of molecular constants, parameters and lines of the red and violet CN system, which could be used as a valuable tool for astrophysical observations. This extensive work summarizes up-to-date spectral data of CN radicals and describes their application in astrophysical measurements.

A special astrophysical application of CN radical spectra lies in the field of outer space temperature estimation. This possibility was found shortly after the estimation of cosmic radiation background temperature (3 K). The temperature of outer space was measured by the use of the rotational energy levels of CN. This method can be seen, for example, in the work of P. Thaddeus (1972) [196].

### 5.2. Experimental Results

The CN radical was observed in the glow discharge of gaseous cyanogen mixed with a helium buffer or formamide–water–nitrogen mixture. In the case of the cyanogen mixture, the partial pressure of gases was 50 mTorr of cyanogen and 3 Torr of helium, and in the case of the formamide mixture, the partial pressure of all components was 1 Torr (formamide, water and nitrogen in 1:1:1 mixing ratio). An InSb semiconductor detector and CaF_2_ lens and beamsplitter were used in all measurements. Spectral resolution varied between 0.07 and 0.025 cm^−1^. The number of scans was 30–50. The pulse width was 20–40 µs; offset 0 µs; data acquisition was carried out at each third microsecond. The electric voltage between the electrodes varied around 1 kV, and the electric current was 50–100 mA.

Figure 22 captures a rotational-vibrational spectrum of CN radical in the 1800–2200 cm^−1^ spectral region obtained by use of a cyanogen mixture, with 50 accumulating scans and 0.027 cm^−1^ spectral resolution. The electric current was 50 mA, and the voltage was 0.7 kV. While the pulse width was set to 20 µs, the spectrum in Figure 22 corresponds to the time of 39 µs (19 µs after the discharge pulse). Other conditions were the same as the general ones.

A comprehensive spectrum of the CN radical in the 2150–3250 cm^−1^ spectral range, demonstrating A^2^Π → X^2^Σ^+^ transitions, is depicted in Figure 23, where a total of 6 bands were measured. The spectrum was obtained under the same conditions as the spectrum in Figure 22.

A sequence of the Δv = −2 of A^2^Π → X^2^Σ^+^ transition type of the CN radical, lying in the 4350–5150 cm^−1^ spectral range, is depicted in Figure 24 and was obtained by the use of (CN)_2_, with a pulse width of 22 and 14 µs after the pulse ended. The spectral resolution was 0.05 cm^−1^, and 50 scans were accumulated to obtain a reasonable signal-to-noise ratio. The electric voltage was 1.4 kV, and the electric current was 100 mA.

Figure 25 shows the spectrum of the CN radical in the spectral region of 6400–7100 cm^−1^, where the Δv = −1 sequence is located and was obtained by the use of a cyanogen-helium mixture.

The Appendix A of this work provide the following spectroscopic data on the CN reported here: the fundamental X^2^Σ^+^ ground state; electronic A^2^Π → X^2^Σ^+^ transition of CN radical for the Δv = −3 sequence, Δ*v* = −2 sequence and Δ*v* = −1 sequence. All lines were assigned according to known spectroscopic notation.

All measured bands of CN radical in the 1800–7500 cm^−1^ spectral region are illustrated in Figure 26, sorted by their overall appearance in the broad-range spectrum.

## 6. CH Radical

### 6.1. Theoretical Background

The spectra of CH radical could be observed at any time when the decay of hydrocarbons during a high-energy event happened. Similar to other radicals, CH is also present in the spectra of products of exothermic chemical reactions, burnings and explosions. The CH radical is present not only in solar spectra but also in interstellar space (e.g., diffuse molecular clouds) and on stars. Alongside OH, NH and CN, CH is considered to be one of the most astrophysically important radicals.

Spectroscopic features of the CH radical were defined by the shape and mutual position of potential functions for specific energy levels (see Figure 27).

The first detection of the CH radical dates back to 1919 by Heurlingher and Hulthen [198]. The spectra of CH were subsequently measured under various conditions for another 50 years. In 1969, Herzberg and Johns [197] summarized the existing data on the CH radical spectra and added new electronic transitions in the regions around 73,000, 65,000, 64,000, 59,000 and 33,000 cm^−1^. In 1978, Filseth et al. [199] measured the fluorescence of CH in low-pressure flames of acetylene mixed with oxygen and hydrogen. He measured the chemiluminescence of the 0-0 band of the A^2^∆ → X^2^Π transition, which is a dominant spectral feature of hydrocarbon combustion. The first publication on the measurement of CH by means of laser magnetic resonance was in 1978. The work by Hougen [200] reported precise data for energies of rotational energy levels of CN in the region of FIR to microwaves while also being among the very first ones to demonstrate the hyperfine splitting of several CH lines. Suzuki et al. [201] measured the emission transitions of CH in the discharge of methane and argon. In 1986 P. Chen et al. [202] studied unknown lines of the CH radical emerging from two-photon transitions to Rydberg and other highly excited valence states. These new bands were found to be around 50,000 cm^−1^ (UV region) and were studied during flash photolysis of ketene, prepared by the flash vacuum pyrolysis of acetic anhydride. P. Bernath et al. [203] observed new emission transitions of CH radical of the A^2^∆ → X^2^Π and B^2^Σ^−^ → X^2^Π type. Spectra of oxygen-acetylene flame were also of interest to Hung et al. [204], who used a modern detection method based on two dye lasers and demonstrated the efficiency of such an experimental arrangement in the detection of radicals. The next 20 years brought many newly observed transitions and energy levels of CH, as described, e.g., in the work of Bembenek et al. (1997) [205] containing measurement of the C^2^Σ^+^ → X^2^Π band of ^13^CH; the work of M. Zachwieja [206] on the A^2^∆ → X^2^Π band was also considered for ^13^CH; the resonant multiphoton ionization spectroscopy of the A^2^∆ → D^2^Π band, ref. [207] revealed a new Rydberg state around 63,000 cm^−1^; submillimeter measurements of CH around 530 GHz (17 cm^−1^) of Amano; ref. [208] similar measurements of Davidson (2001) [209] in the FIR region, with a high measurement precision of only 100 kHz; the interesting work of Czyzewski et al. (2002) [210] about the decay kinetics of CH, measured by means of the cavity ring-down spectroscopy; measurements of ^13^CH in the FIR region by the use of aforementioned laser magnetic resonance; ref. [211] measurements of the first vibrationally excited (*v* = 1) energy level of X^2^Π state by Jackson et al. (2008) [212]; measurement of CH lines of extreme frequencies (535 GHz) with high resolution by S. Truppe et al. (2014) [213] providing possible information about the stability of physical constants; or the modern work by Gans et al. (2016) [214], which focused on an application of synchrotron radiation in the study of one-photon transitions of CH of X^2^Π → X^+1^Σ^+^ or X^2^Π → a^+3^Π type.

CH radical cation (CH^+^) was also of interest to many extensive researchers. Namely, the works of M. Carre (1969) [215] or Carrington and Ramsay (1982) [216] can be mentioned.

Many theoretical works were dedicated to spectral features of the CH radical. The radiation lifetime of A^2^∆ was analyzed by Carozza and Anderson (1977) [217] and was found to be 508 ± 25 for $v^{\prime }$ = 0 ns according to the available data. The work of Jørgenssen et al. [218] from 1996 demonstrated an extensive database of existing CH lines, constructed through the use of the modeling of quantum-allowed energy transitions between X^2^Π → X^2^Π, A^2^∆ → X^2^Π, B^2^Σ^−^ → X^2^Π and also C^2^Σ^+^ → X^2^Π. By using this model, an open-access online database of 112,821 lines of CH was created. J. Martin [219] published new potential functions of the CH radical, using ab initio calculations to obtain the accurate values of energy level positions and bond lengths for CH, NH, OH and HF with only 1 cm^−1^ of uncertainty. A. Kalemos et al. [220] theoretically investigated Rydberg states of the CH radical and provided a set of accurate spectroscopic constants. The vibrational transition moment of the CH radical was examined by Ghosh et al. (1999) [221], who obtained this parameter according to a defect of the P and R branch intensity (so-called Herman-Wallis effect). A. Metropoulos et al. [222] studied highly excited states of the E^2^Π and F^2^Π type and provided a pre-dissociative lifetime for the E^2^Π state of 2 ps. From other theoretical works, there was the publication by Heryadi et al. (2002) [223] on a direct determination of ionization potentials of CH; the work by Reddy et al. (2004) [224] on another precise calculation of potential functions of CH and the subsequent calculation of the dissociation energy of this radical (3.48 ± 0.90 eV); the ab initio work by Vázquez et al. (2007) [225] on the Rydberg states of CH; the publication by Lavín et al. (2009) [226] about oscillator strengths and their spectral distribution of CH; or a work by Masseron et al. (2014) [227], where all available positions and intensity distributions of the lines and bands of CH radical and their presence in stellar atmospheres were exhaustively described.

CH was one of the first species to be spectrally detected in astronomical sources of electromagnetic radiation. Firstly, it was detected in diffuse interstellar clouds by Swings and Rosenfeld in 1937 [228]. Later CH was identified in cometary spectra (Swings and Nicolet, 1938) [229] and solar spectra (Wildt, 1941) [230]. The lines first observed on stars or in interstellar space were assigned later, e.g., by Douglas and Morton [231] in 1960 or by Watson in 2001 [232].

Mélen et al. [233] also investigated solar spectra in 1989, where they used data acquired by the ATMOS probe. New 558 spectral lines of the CH radical were identified in the solar spectra. Data from the ATMOS mission were also investigated by Zachwieja (1995) [234], who assigned another 144 new lines of CH.

High-quality CH referential spectra allow a study of exoplanetary atmospheres by the estimation of carbon and hydrogen abundance. The CH spectra can also be used to determine the ^12^C/^13^C ratio, uncovering the evolution path of a specific star.

### 6.2. Experimental Results

CH radical was spectrally examined in the glow discharge of methane mixed with helium as a buffer gas. The partial pressure of helium was 3 Torr, and the partial pressure of methane was 0.05 Torr. An InSb detector and CaF_2_ optics were used. An unapodized spectral resolution was set to 0.035 cm^−1^, and 64 scans were accumulated. Two experimental assignments were used with a pulse width of 10 and 40 µs and an offset of 3 and 35 µs. Data acquisition was set to each 1st µs, the electric voltage was 1.2 kV, and the electric current was 100 mA.

An overall spectrum of the glow discharge of the aforementioned mixture is illustrated in Figure 28, which describes the expected MIR spectrum, containing minor products as well. As can be seen in this figure, the spectrum contains several minor species such as molecular hydrogen (5g-4f and 2p-2s transitions), acetylene (ν_3_), and the C_2_ radical (B^1^Δ_g_ → A^1^Π_u_ electronic transition).

Emission lines of the CH radical from Figure 28 represent rotational-vibrational transitions in the X^2^Π ground state and are associated into quadruplets. A detail of one of those quadruplets is shown in Figure 29.

All emission CH radical lines reported here are provided in the Appendix A with the full assignment. The assignment of individual transitions was performed using the ExoMol [235] data, which was derived from the MoLLIST [156] database.

### 6.3. Comparison with the Solar Spectra

A CH radical was found, among many other species, in the solar ACE spectra, with the typical structure of the CH radical illustrated in Figure 29, Figure 30 and Figure 31. Figure 31 shows the laboratory-detected CH spectrum in the 2850–2870 cm^−1^ region, together with the solar spectrum taken by the ACE mission. The green highlights in Figure 31 indicate the coincidences with the ACE, namely for the CH *v* = 1 → 0 and v= 2 → 1 vibrational transition. As can be seen, while the laboratory experimental CH spectra for transitions of this type v=1 → 0 were relatively intense, v=2 → 1 transitions were already weak. The solar lines of the CH radical were relatively weak compared to presented atomic lines, such as silicon or iron, but this phenomenon is in accordance with the well-known fact of the generally higher intensity of atomic transitions [236] compared to molecular transitions [236].

## 7. Conclusions

Radicals such as OH, NH and CN, together with excited CO, play a dominant role either in high-temperature shock waves or in discharge chemistry in mixtures containing C, H, O, and N compounds. The CN radical and CO molecule is unique in being able to store a huge amount of thermal energy that less stable species would otherwise dissipate by dissociation and multi–step reactions. The resilience of the CN radical and CO molecule, due to strong triple bonds and the excitation of numerous vibrational and rotational transitions, thereby controls the energy flow in high energy C, H, O and N contains chemical systems (meaning these species are readily formed in such systems and tend to absorb the most energy from external sources). The stable products in all the studied mixtures are HCN, CO, and NH_3_. Subsequent reactions of these species with each other and with CN and other radicals (H and NH_x_, NO [237]) occur to some extent and produce a few larger molecules, some of which are relevant in the context of the origin of life [19].

The foregoing describes the chemistry from a material perspective, but it is also pertinent to consider it from an energy perspective as the reductive homologation chemistry relies on two sources of energy: the UV photons that drive the production of hydrated electrons from certain anions; and the triple bonds of hydrogen cyanide and other nitriles that derive from it and provide the chemical energy to drive subsequent reactions.

Small radicals also play a very important role inside the interstellar medium (ISM). The elemental composition of the ISM is, of course, essential to interstellar chemistry. The ISM contains about 90% of hydrogen (atomic and molecular) and 8–9% of helium, which does not take part in chemical reactions, except that He^+^ intervenes in charge-exchange reactions or neutralization reactions. The remaining 1–2% contains all the other elements, which are, thus, only present as traces. The most abundant are C, N, O and S, Si, Fe.

The field of interstellar small radicals has also stimulated considerable theoretical developments in atomic and molecular physics and physical chemistry, from the ab initio molecular structure to reactive collision studies. No less important experimental set-ups have been built due to the impulse from astrophysicists, from the infrared spectroscopy of radicals and unstable molecules to direct studies of ion-molecule and surface chemistry. Clearly, all these studies have been rewarding for both sides; they provide critical data for astrophysicists, while physicists and chemists have found it in interstellar medium-density conditions, which would never be achievable in terrestrial laboratories. Molecules that are very unstable in laboratory conditions can survive for so long in the interstellar medium that they become observable targets for astronomical research; in turn, their production and study in the laboratory becomes quite challenging.

## Figures and Tables

**Figure 1 molecules-28-03362-f001:**
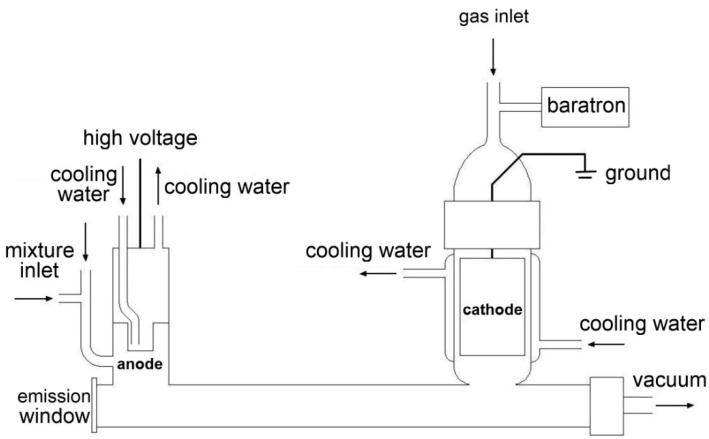
Scheme of the discharge cell.

**Figure 2 molecules-28-03362-f002:**
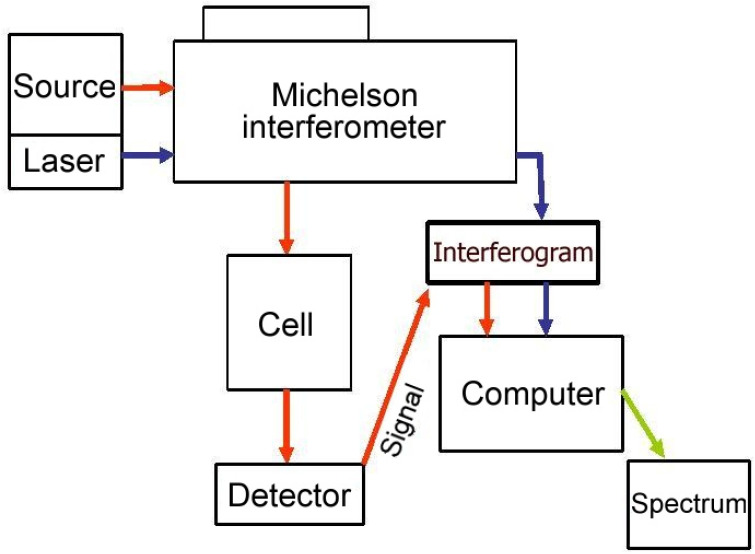
Inner arrangement of Bruker IFS 120 HR spectrometer [90].

**Figure 3 molecules-28-03362-f003:**
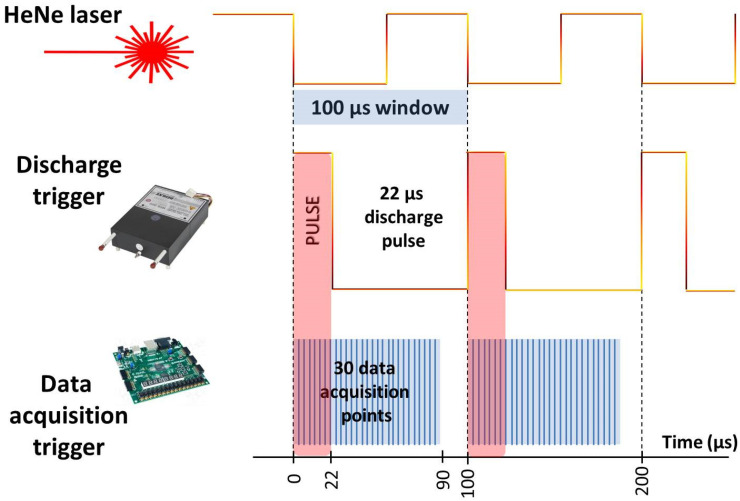
Time-resolved data acquisition setup.

**Figure 4 molecules-28-03362-f004:**
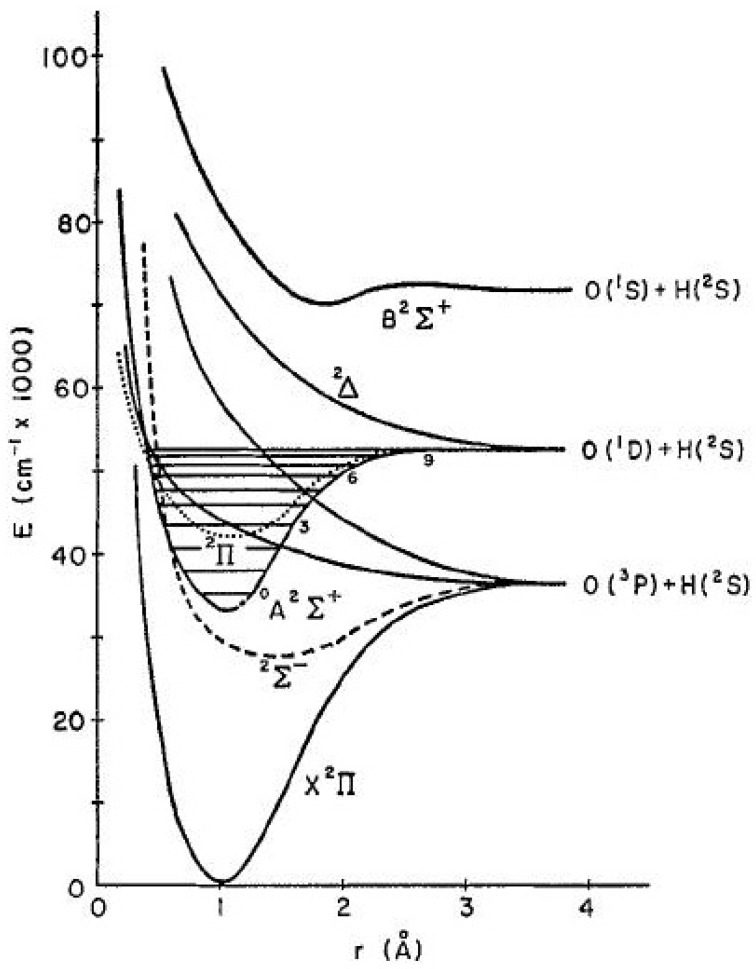
Potential curves of OH radical for low energy states [91].

**Figure 5 molecules-28-03362-f005:**
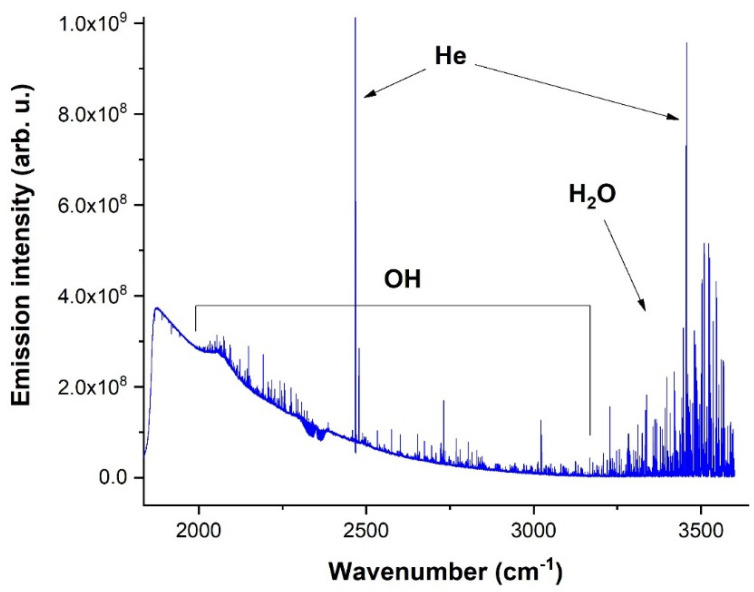
Averaged spectrum of OH radical in the 1800–3500 cm^−1^ spectral region.

**Figure 6 molecules-28-03362-f006:**
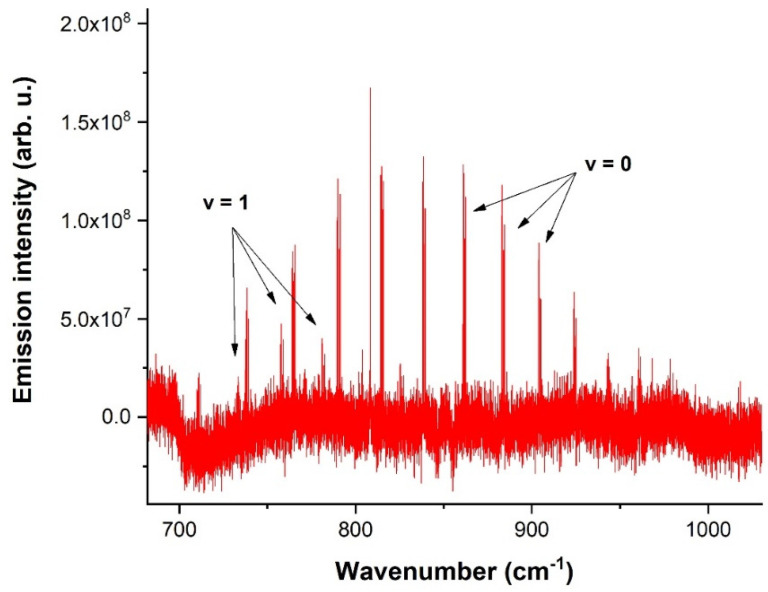
Rotational spectrum of OH radical in the 700–1000 cm^−1^ spectral range in the ground (*v* = 0) and first (*v* = 1) vibrationally excited state.

**Figure 7 molecules-28-03362-f007:**
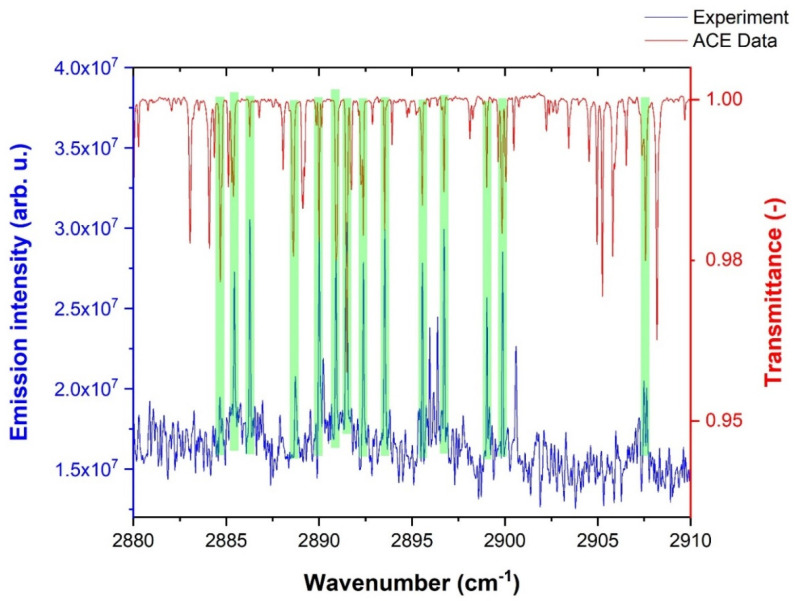
Comparison of experimentally observed rovibrational OH emission lines with the ACE solar spectrum.

**Figure 8 molecules-28-03362-f008:**
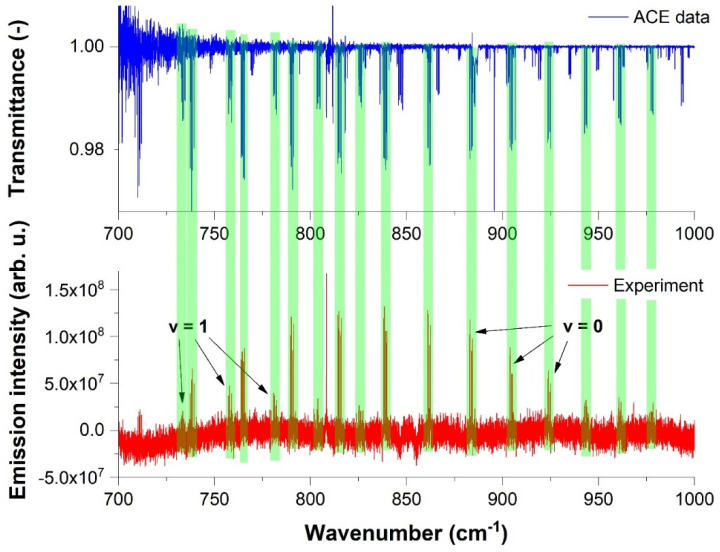
Comparison of the experimental OH spectrum with ACE solar spectrum in the 700–1000 cm^−1^ spectral range.

**Figure 9 molecules-28-03362-f009:**
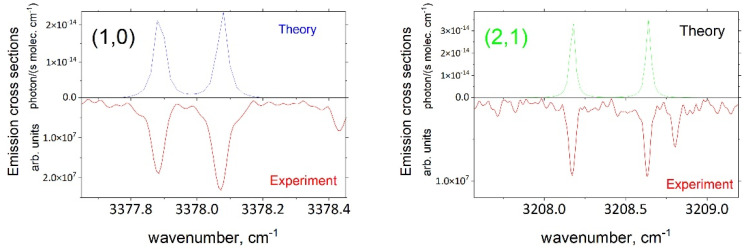
Theoretical emission spectra of OH (1,0 and 2,1) simulated using the MoLLIST line list ($T_{\mathrm{rot}}=$500 K and $T_{\mathrm{vib}}=$ 3620 K, using a Lorentzian line profile of HMHM = 0.02 cm^−1^) with the experimental spectrum from this work.

**Figure 10 molecules-28-03362-f010:**
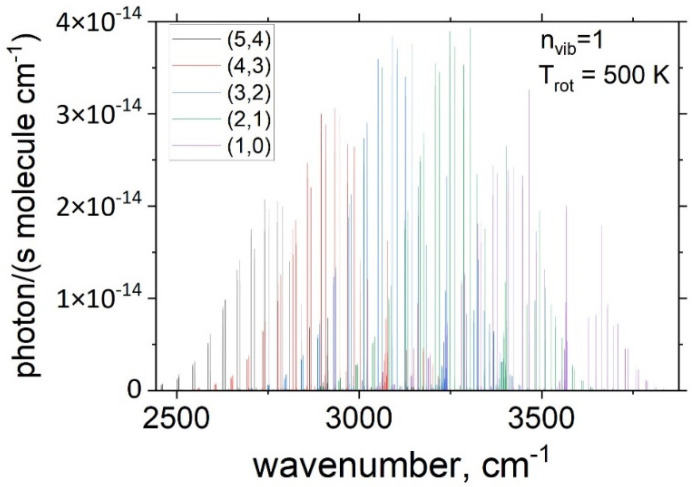
Theoretical emission spectra (cross sections) of OH for the 5 bands simulated using Equation (1) ($T_{R}=$ 500 K, using a Lorentzian line profile of HMHM = 0.02 cm^−1^) assuming the vibrational population $n_{v}=1$.

**Figure 11 molecules-28-03362-f011:**
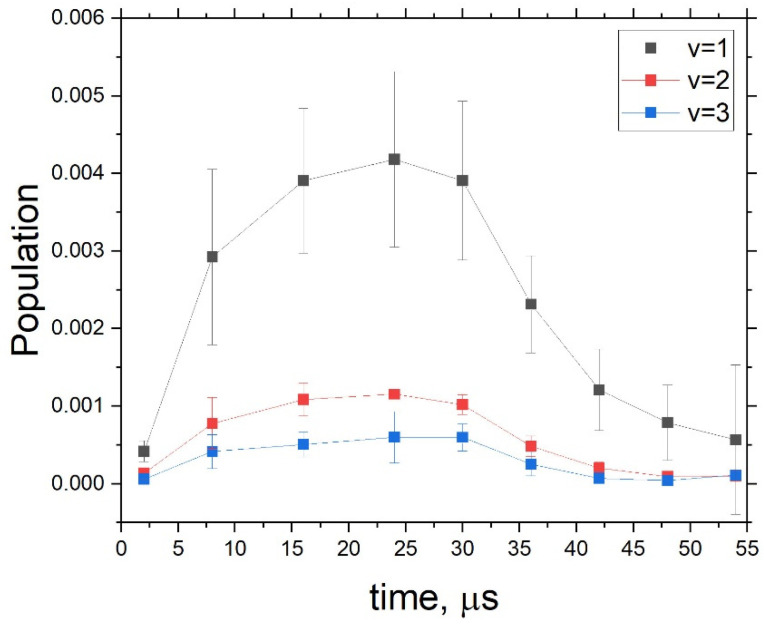
Vibrational populations of OH four vibrational states calibrated to the theoretical spectra using Equation (6) (assuming $T_{R}=$ 500 K).

**Figure 12 molecules-28-03362-f012:**
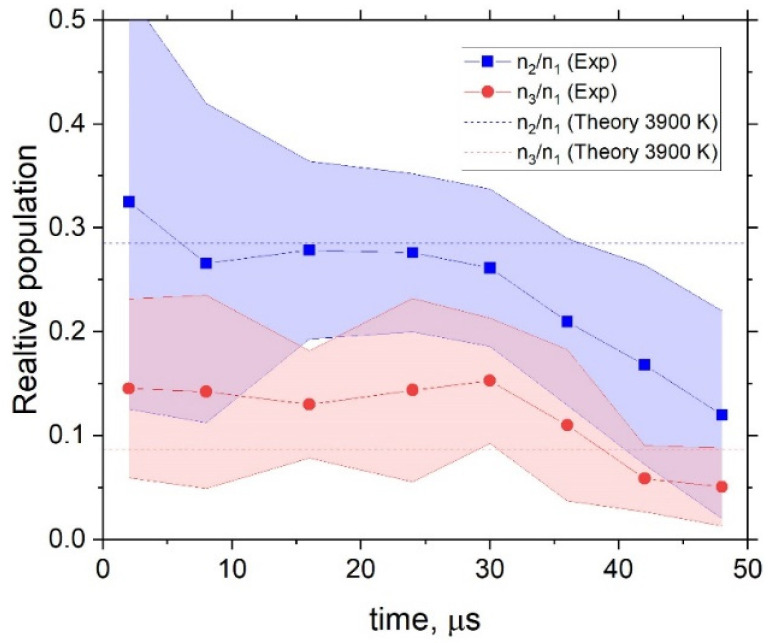
Populations of OH of v=2 and v=3 relative to the population of v=1, assuming $T_{R}=$ 500 K. The dash lines indicate the estimated relative populations for the vibrational temperature of $T_{\mathrm{vib}}$ = 3900 K.

**Figure 13 molecules-28-03362-f013:**
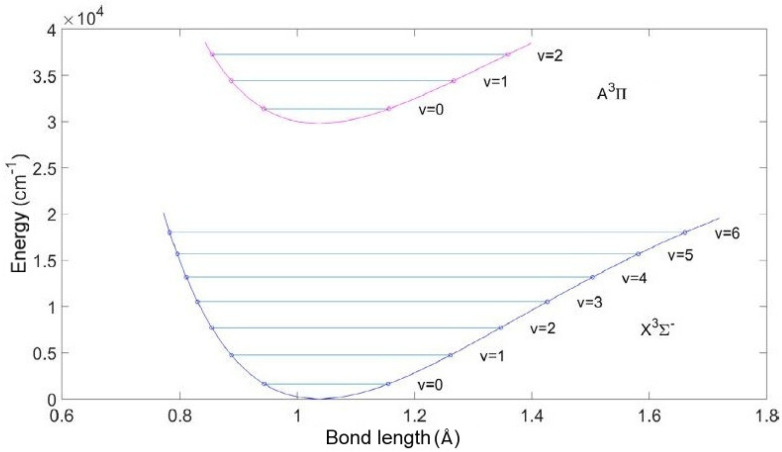
Potential curves for low-lying energy levels of NH [125].

**Figure 14 molecules-28-03362-f014:**
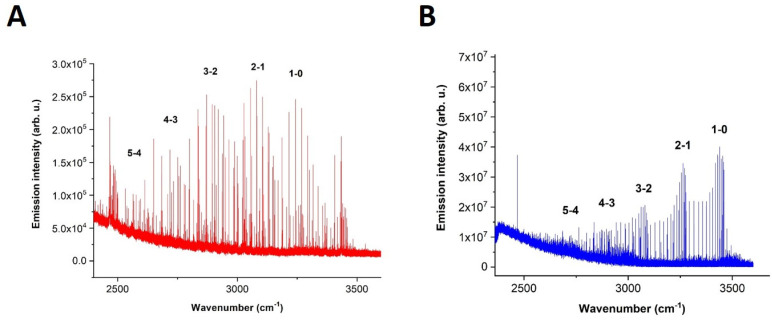
Panel (**A**)—averaged spectrum of H_2_ + N_2_ discharge. Panel (**B**)—averaged spectrum of NH_3_ + N_2_ + Ar discharge.

**Figure 15 molecules-28-03362-f015:**
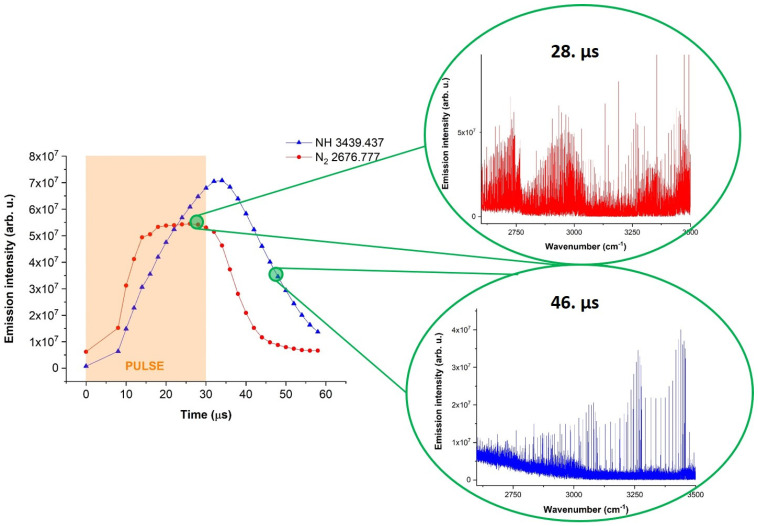
Time profiles of NH radical and N_2_ and the appearance of spectra at specific times.

**Figure 16 molecules-28-03362-f016:**
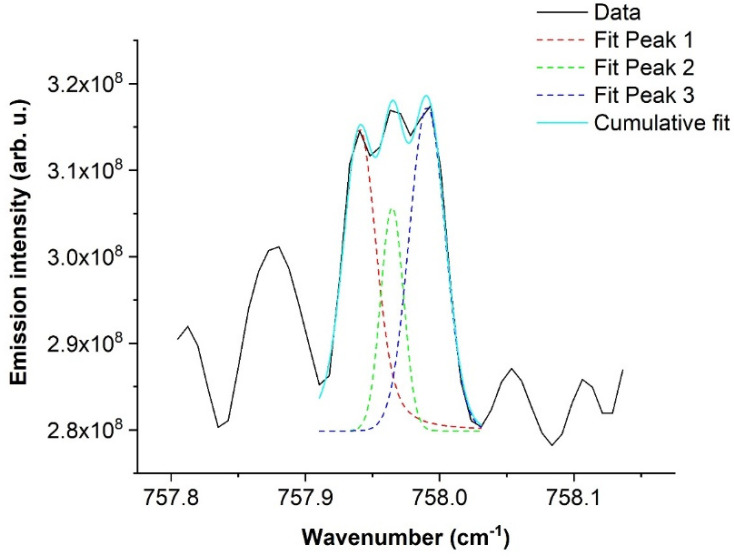
Deconvolution of an NH radical triplet line ($J^{\prime \prime }$ = 26, v = 0) [153].

**Figure 17 molecules-28-03362-f017:**
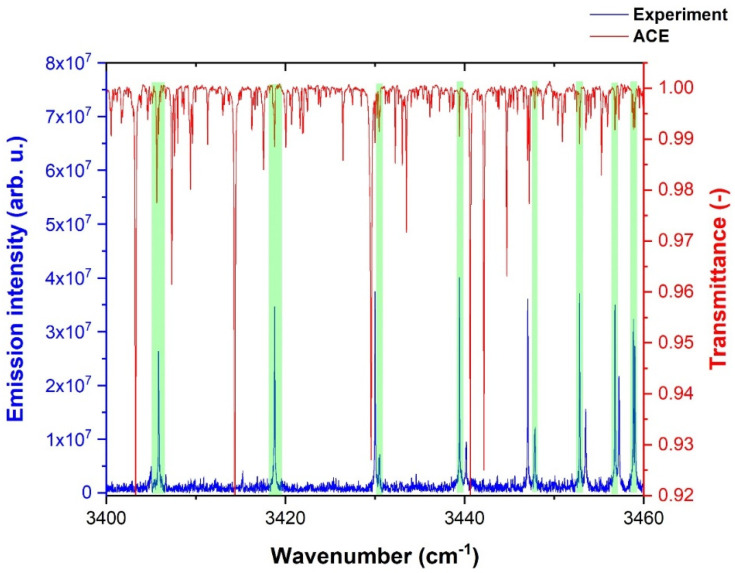
Comparison of ACE solar spectrum and our NH radical data [153].

**Figure 18 molecules-28-03362-f018:**
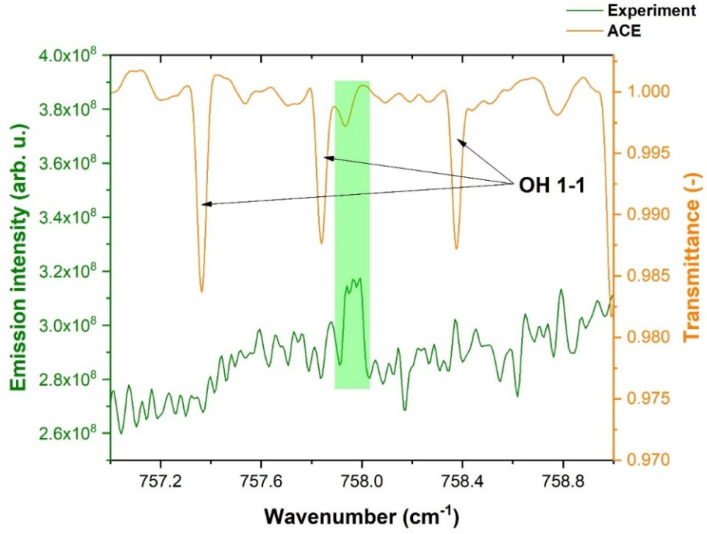
The comparison of ACE solar spectrum with our NH pure rotational triplet [153].

**Figure 19 molecules-28-03362-f019:**
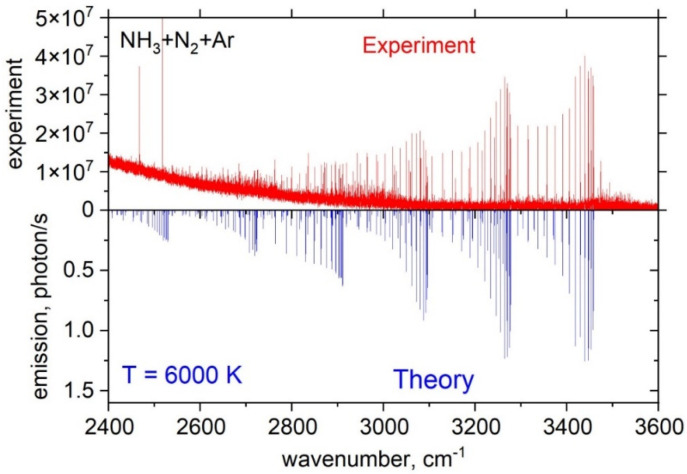
Experiment (red) versus theory (blue) modeling of the NH spectra between 2500 and 3500 cm^−1^ obtained from the NH_3_ + N_2_ + Ar mixture. The theoretical spectrum corresponds to *T*_V_ = 6000 K and *T*_R_ = 500 K, the Lorentzian line profile and a HWHM = 0.05 cm^−1^. The MoLLIST [156,157] line list was used.

**Figure 20 molecules-28-03362-f020:**
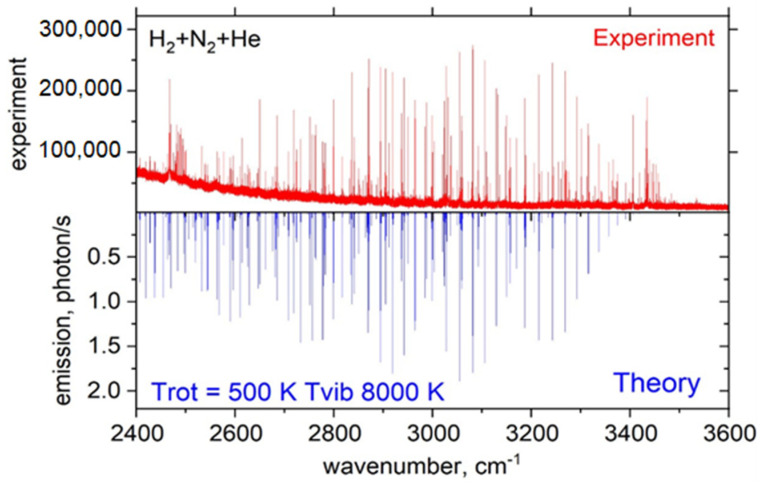
Experiment (red) versus bi-temperature (blue) modeling of the NH spectra between 2400 and 3600 cm^−1^ obtained from the H_2_ + N_2_ + He mixture. The theoretical spectrum corresponds to the non-LTE combination of *T*_R_ = 500 K, *T*_v_ = 8000 cm^−1^, Lorentzian line profile and a HWHM = 0.02 cm^−1^. The MoLLIST [156,157] line list was used.

**Figure 21 molecules-28-03362-f021:**
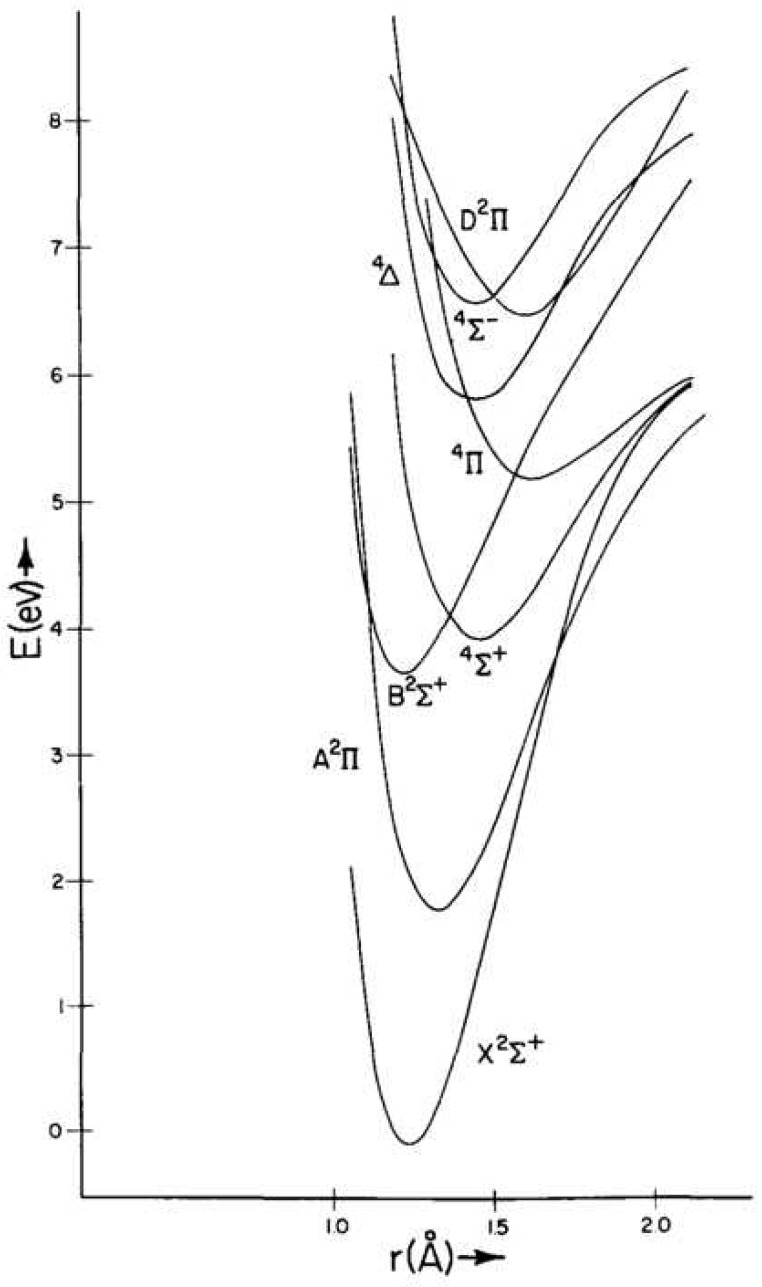
Potential curves [166] for selected low-energy levels of CN.

**Figure 22 molecules-28-03362-f022:**
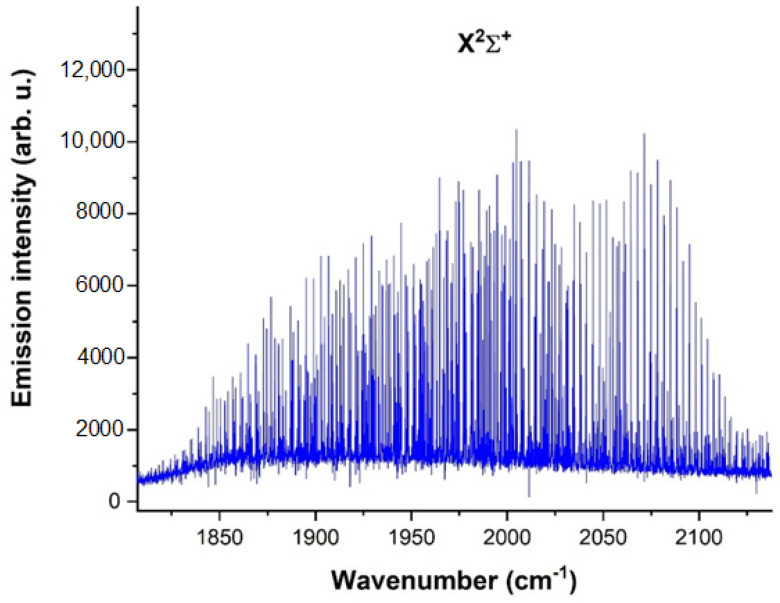
Several rotational-vibrational bands of CN radical in the X^2^Σ^+^ ground state.

**Figure 23 molecules-28-03362-f023:**
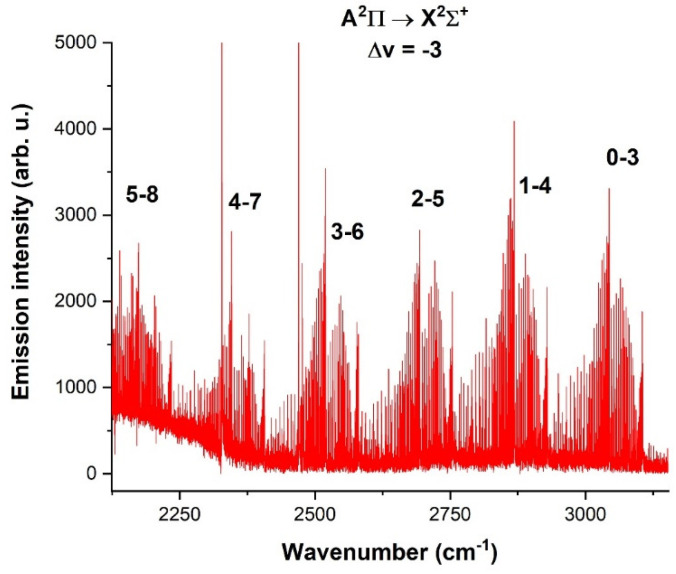
Spectrum of electronic bands of CN radical for the Δ*v* = −3 sequence (39 µs).

**Figure 24 molecules-28-03362-f024:**
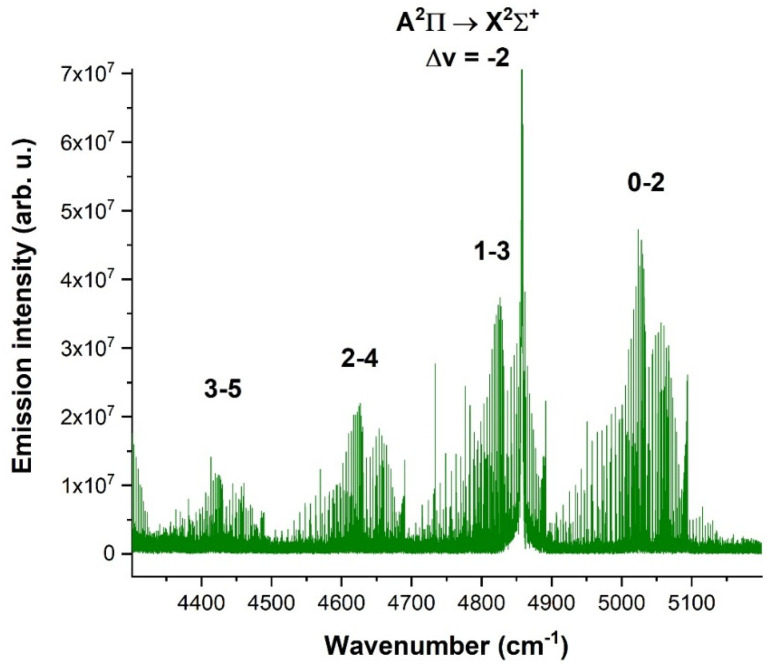
Spectrum of electronic bands of CN radical for the Δ*v* = −2 sequence (36 µs).

**Figure 25 molecules-28-03362-f025:**
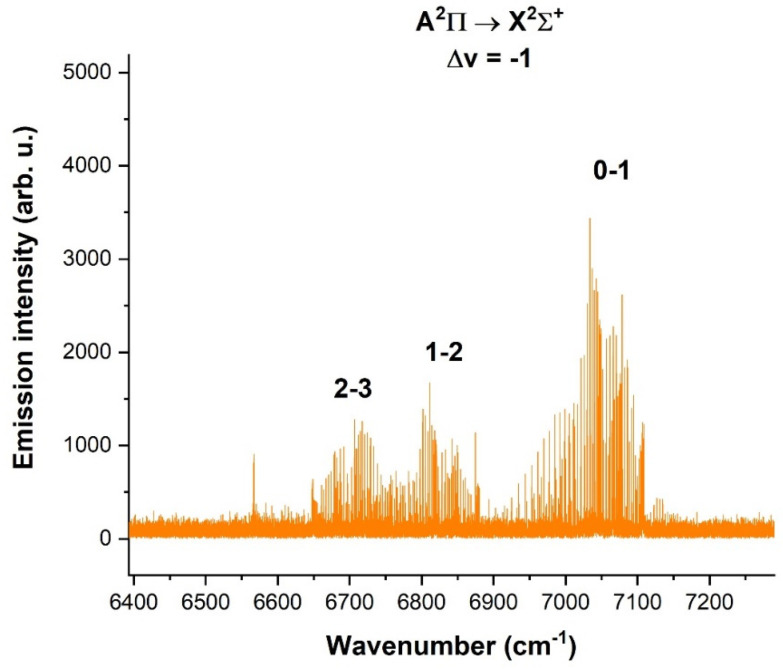
Spectrum of electronic bands of CN radical for the Δ*v* = −1 sequence (24 µs).

**Figure 26 molecules-28-03362-f026:**
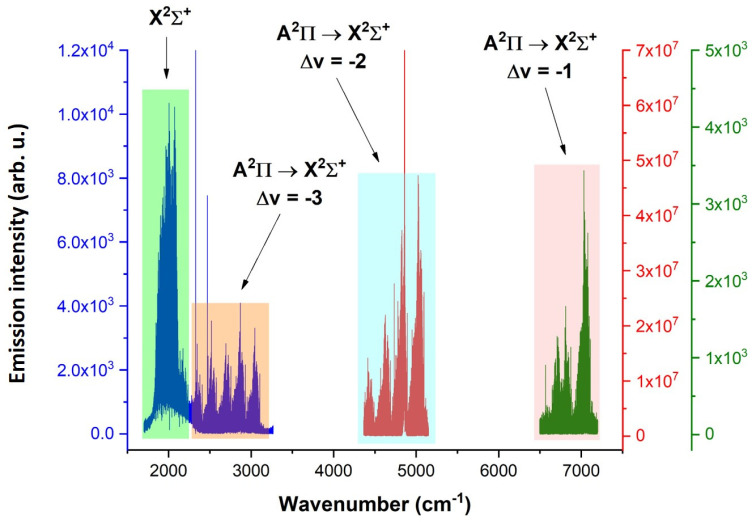
Overall spectrum of all obtained CN bands in the 1800–7500 cm^−1^ spectral range.

**Figure 27 molecules-28-03362-f027:**
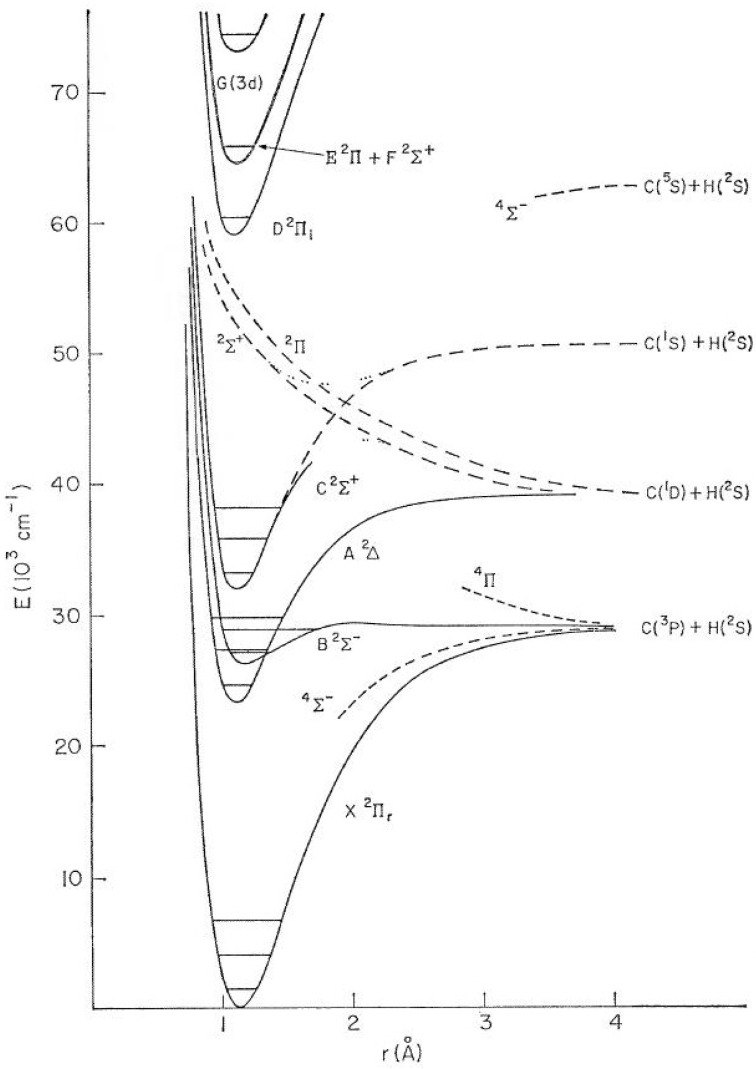
Potential curves of the CH radical for low-lying energy levels [197].

**Figure 28 molecules-28-03362-f028:**
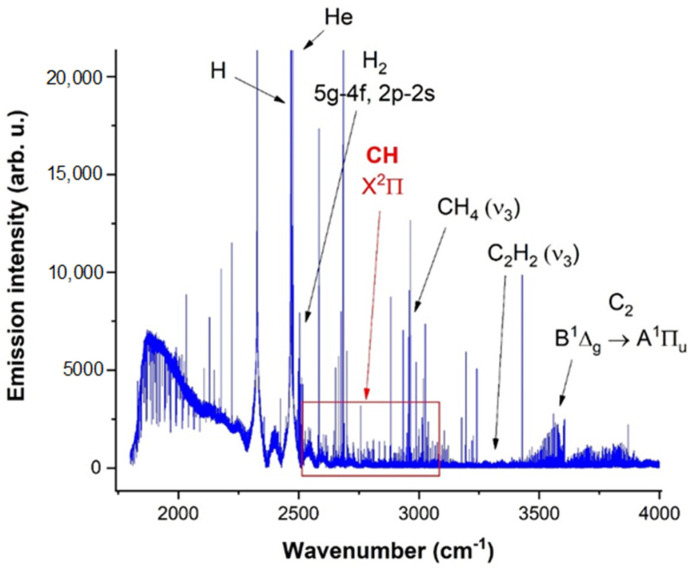
Overall spectrum of the glow discharge of the CH_4_ + He mixture (47 µs).

**Figure 29 molecules-28-03362-f029:**
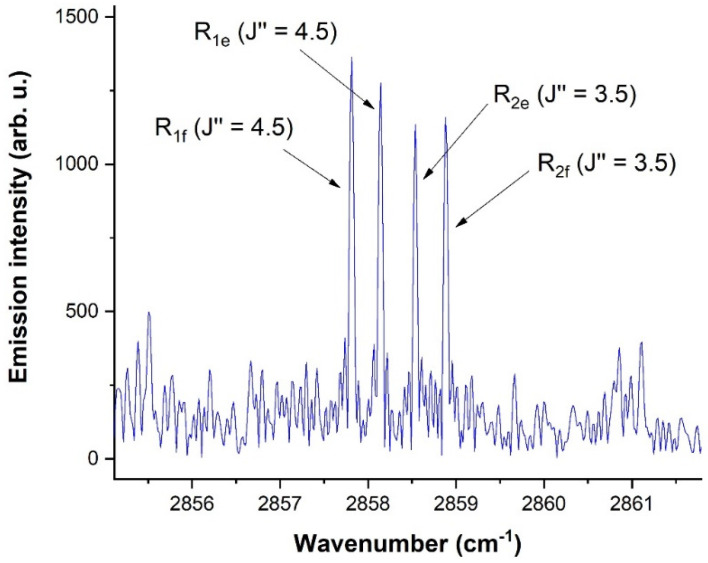
A detailed look at the CH quadruplet around 2858 cm^−1^.

**Figure 30 molecules-28-03362-f030:**
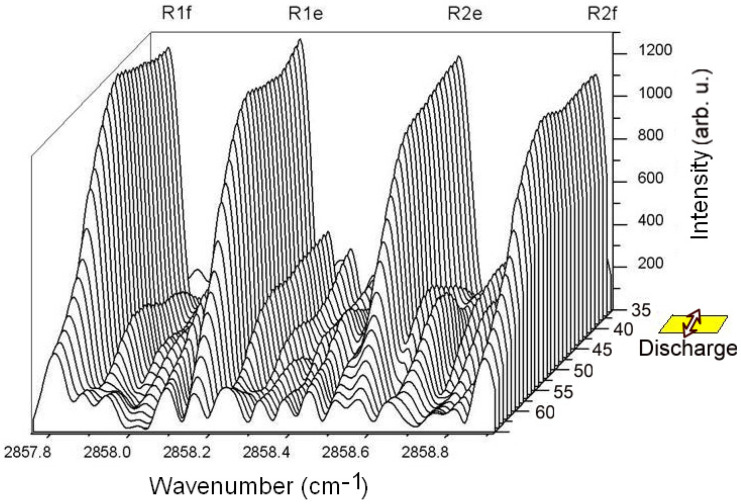
The time profile of CH radical at 2858 cm^−1^. The time interval of the discharge pulse is marked in yellow.

**Figure 31 molecules-28-03362-f031:**
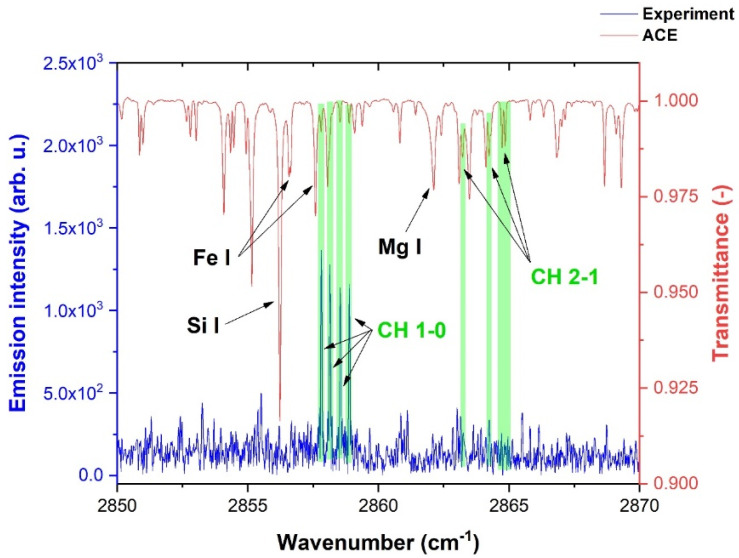
The comparison of ACE solar spectrum and our laboratory CH radical data.

**Table 1 molecules-28-03362-t001:** Overview of the laboratory observed pure rotational NH lines (in cm^−1^).

$J^{\prime \prime }$	Our Position	Solar Position *
26	757.940	757.936
	757.966	757.964
	757.993	757.992
27	776.966	776.966
	776.990	776.986
	777.014	777.009
28	795.104	795.099
	795.125	795.122
	795.146	795.143
29	812.301	812.310
	812.329	812.332
	812.355	812.360

* Solar wavenumbers in the Table were extracted from ATMOS (Atmospheric Trace Molecule Spectroscopy) experiment (29 April–2 May 1985), as described and identified in publications by Farmer and Norton [154] and Geller et al. [155]. For each $J^{\prime \prime }$ number a triplet of NH lines is described in the Table of Pastorek et al. [153].

## Data Availability

Data available upon request.

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
