# Peer review of "Infrared Spectra of Small Radicals for Exoplanetary Spectroscopy: OH, NH, CN and CH: The State of Current Knowledge"

_molecules, 2023, doi:10.3390/molecules28083362_

Round 1

Reviewer 1 Report

The present manuscript reports on the infrared emission spectra of some diatomic free radicals: OH, NH, CN, CH. Some transitions for these molecules are measured by FTIR spectroscopy at medium resolution, namely between 0.07 and 0.02 cm-1. Given the astrochemical importance of such species, comparisons with experimental solar spectra or simulations based on the MoLList are often displayed.

The manuscript is interesting, as it summarizes a great deal of the work that has been done previously, with a large bibliography. It also provides new experimental data that might help the detection of the radical in remote environments, in particular the exoplanetary atmospheres.

II believe that this manuscript deserves to be published in this journal without major changes. However, I recommend some minor changes that, I hope, will improve its overall clarity.

Page 6, line 15. "In" is repeated twice.

Page 8, line 24. Abundance was "the" work of… (add the article)

Page 10, line 5. Which of your transitions for OH are already in HITRAN? Have you detected new transitions? Please clarify.

Figures 5, 6, 7, 8, 14, 15, 16, 17, 18, 22, 23, 24, 25, 26, 28, 29, 31. The y axis title is "Emission intensity (-)". When the units are not defined it would be better to write "Emission Intensity (arbitrary units)" which, by the way, you report in other figures.

Fig. 19. There’s quite a difference between the experimental and theoretical spectra, above all between 2400 and 2900 cm-1. Please add a comment about it.

Supp. Mat. The tables reported in the Supplementary Material should have an explanation (maybe as footnotes) of the content of each column. nu_exp and nu_the may sound self-explanatory but what is nu? Also, since no equations are reported in the main text, I would also explain what the quantum numbers J, N and v are.

Reviewer 2 Report

The paper presents an excellent review of the current state of the art in the spectroscopy of diatomic radicals in a close relation with many astrophysical applications. It is well structured and well written with nice general introduction of the needs in these data for exoplanetary physics and chemistry. Each chapter devoted to the original studies of the four radicals is preceded by an in-depth historical overview.

This valuable paper is certainly of interest for a large community of the readers. I strongly recommend it for publication in “MOLECUES” MDPI  with the following minor comments to be accounted for:

1.      Low technical quality of the copy/paste Figures 4,13,16,21,23,24,30 must be improved at the final ready-to-publish version.

2.      Specify the emission intensity units in Figures 5,6,14,22

3.      Specify the quantity at the vertical scale plotted in Fig 10

4.      Figure 15 must be commented in more detail:

-         specify the meaning of numbers in NH 3439.437, N2 2676.777

-         does this mean that the red plot spectrum at 28th mu_s corresponds to the N2 emission ?

-         if yes this looks surprisingly crowded for a diatomic species without dipole moment; What data sources were used to identify the spectrum ?

5.      Fig19 and Fig 20 : Is there a plausible explanation of the missing patterns in the experimental spectrum at 2400- 2700 cm-1 ?

Would this mean that the theoretical intensities are too strong or is there lack of a correct non-LTE modeling ?   Please add a comment.

6.      For the completeness of the review the following references are to be added

R. Gamache et al , Partition sums for non-local thermodynamic equilibrium conditions for nine molecules of importance in planetary atmospheres, Icarus 378 (2022) 114947, https://doi.org/10.1016/j.icarus.2022.114947,

which describes the non-LTE procedure of calculations including OH and NO,

as well as missing refs

for OH data

J. Lelieveld, S. Gromov, A. Pozzer and D. Taraborelli, Atmos.Chem.Phys. 16, 12477 (2016). doi:10.5194/acp-16-12477-2016

M.A.Martin-Drumel, O. Pirali, D. Balcon, P. Brechignac,P. Roy and M. Vervloet, Rev. Sci. Inst. 82, 113106 (2011).doi:10.1063/1.3660809

 S. Noll, H. Winkler, O. Goussev and B. Proxauf, Atmos.Chem. Phys. 20 (9), 5269 (2020). doi:10.5194/acp-20-5269-2020

O. Sulakshina and Yu. Borkov Mol.Phys , 2018, VOL. 116, NOS. 23–24, 3519–3529

and for NO data

Lee Y-P , Cheah S-L , Ogilvie JF .. Infrar Phys Technol 2006;47:227–39 .

O. Sulakshina , Yu. Borkov JQSRT 209 (2018) 171–179

7.      A comment must be added to describe the content of the Supplementary file:

What is the source for the theoretical data ? Are the experimental line positions original or compiled from various sources ?  What is an estimated accuracy for measured line positions ?

Reviewer 3 Report

The authors have presented a review article entitled“Infrared Spectra of Small Radicals for Exoplanetary Spectroscopy: OH, NH, CN, and CH. The State of Current Knowledge”.

The present review contains  appropriate piece information for readers in the field of Exoplanetary Spectroscopy however, a few queries and revisions must be addressed to enhances the weightage of the review before  its publication.

Abstract:

1.      Why authors are only focused on OH, NH, CN, and CH molecular radicals why not others one?

2.      Can the authors  explain the difference between simple FTIR and -time-resolved FTIR? What is the advantage over a simple one?

3.      What does it mean specially designed discharge cell? How it is designed.

4.      How it helps to analyse the radicals like OH, NH, CN, and CH molecules using the James Webb telescope and Plato and Ariel satellites for the analysis of these radicals.

 Introduction :

1.      How you can distinguish the atmospheric spectral imprints of different molecular radicals i.e. OH, NH, CN, and CH with the help of JWST and Ariel. Is there any standard data available to identify  such radicals?

2.      What is Lunar crater density how is it calculated?

3.      What is the meaning of  an exo-atmosphere means

 Instrumentation of the spectral measurements:

1.      What is the role of discharge trigger and data acquisition trigger in time-resolved Fourier transform infrared?

2.      Why you fixed the mobile mirror speed frequency at 10 KHz, what was the reason why not you have taken  other frequencies?

3.      What kinds of samples and how it prepared before recording  the spectra of different radicals?

Results

1.      Use the abbreviations at least once throughout the paper.

2.      What does the significant role to determine and comparing the theoretical and experimental non-LTE conditions throughout the paper?

3.       In fig. 26 where is the Y axis of the purple colour CN band emission?

4.      What is the meaning of Emission Intensity (-) in the Y axis what are its units?

Conclusion :

1.      How you say that energy flow is controlled using C, H, O & N containing chemical system.

Reviewer 4 Report

I think this review can be accepted for publication in Molecules as it is. 
